# Activation of PKA via asymmetric allosteric coupling of structurally conserved cyclic nucleotide binding domains

Yuxin Hao [1], Jeneffer P. England[1], Luca Bellucci[2], Emanuele Paci[3], H. Courtney Hodges [4,5,6,7], Susan S. Taylor [8,9] & Rodrigo A. Maillard [1]

Cyclic nucleotide-binding (CNB) domains allosterically regulate the activity of proteins with diverse functions, but the mechanisms that enable the cyclic nucleotide-binding signal to regulate distant domains are not well understood. Here we use optical tweezers and molecular dynamics to dissect changes in folding energy landscape associated with cAMP-binding signals transduced between the two CNB domains of protein kinase A (PKA). We find that the response of the energy landscape upon cAMP binding is domain specific, resulting in unique but mutually coordinated tasks: one CNB domain initiates cAMP binding and cooperativity, whereas the other triggers inter-domain interactions that promote the active conformation. Inter-domain interactions occur in a stepwise manner, beginning in intermediate-liganded states between apo and cAMP-bound domains. Moreover, we identify a cAMP-responsive switch, the N3A motif, whose conformation and stability depend on cAMP occupancy. This switch serves as a signaling hub, amplifying cAMP-binding signals during PKA activation.

[1] Department of Chemistry, Georgetown University, Washington, DC 20057, USA. [2] NEST, Istituto Nanoscienze del CNR and Scuola Normale Superiore, Pisa 56127, Italy. [3] Astbury Centre & School of Molecular and Cellular Biology, University of Leeds, Leeds LS2 9JT, UK. [4] Department of Molecular and Cellular Biology and Center for Precision Environmental Health, Baylor College of Medicine, Houston, Texas 77030, USA. [5] Dan L Duncan Comprehensive Cancer Center, Baylor College of Medicine, Houston, Texas 77030, USA. [6] Center for Cancer Epigenetics, The University of Texas MD Anderson Cancer Center, Houston, Texas 77030, USA. [7] Department of Bioengineering, Rice University, Houston, Texas 77005, USA. [8] Department of Pharmacology, University of California, San Diego, La Jolla, California 92093, USA. [9] Department of Chemistry and Biochemistry, University of California, San Diego, La Jolla, California 92093, USA. Correspondence and requests for materials should be addressed to R.A.M. (email: rodrigo.maillard@georgetown.edu)

Throughout evolution, nature has utilized structurally conserved protein domains as regulatory signaling modules[1–6]. In multi-domain assemblies, these signaling modules communicate and transduce ligand-binding signals to other functional domains, thereby enabling diverse responses to intracellular signaling cascades[7,8]. Cyclic nucleotide-binding (CNB) domains are ubiquitous and structurally conserved signaling modules that regulate the activities of protein kinases, guanine nucleotide-exchange factors, nucleotide-gated channels, and transcription factors in response to cyclic nucleotides[1]. To date, a general understanding of how the activity of CNB domains can be adapted to regulate a diverse array of protein functions remains rudimentary.

Here we used optical tweezers in combination with steered molecular dynamic (SMD) simulations to study the mechanisms that link cyclic adenosine monophosphate (cAMP) binding and inter-domain communication with allosteric regulation of cAMP-dependent protein kinase A (PKA). PKA is an archetype of cyclic nucleotide-dependent protein kinases that is composed of regulatory and catalytic subunits[9]. The phosphorylating activity of the catalytic subunit is allosterically driven by two CNB domains of the regulatory subunit, termed CNB-A and CNB-B[10–12]. cAMP binding starts in the CNB-B domain and enables binding of a second cAMP molecule to the CNB-A domain, resulting in a profound conformational change that unleashes the activity of catalytic subunits[10].

Our studies show that cAMP binding to the two CNB domains of PKA propagates a reorganization of inter-domain contact nodes that reshape the folding energy landscape of the protein. Changes in the energy landscape are unique to each CNB domain and arise from both ligand-binding and inter-domain interactions. We identify a division of labor among CNB domains: the CNB-B domain is responsible for initiating and triggering cAMP binding cooperativity, whereas the CNB-A domain induces strong inter-domain interactions that lock the entire protein complex into its active conformation. Moreover, we identify a cAMP-responsive structural element, the N3A motif, which switches in stability and conformation depending on cAMP occupancy and inter-domain contacts. Through mutagenesis and the use of cyclic nucleotide analogs, we show that this ligand-responsive switch is selective to cAMP and serves as a signaling hub, amplifying the cAMP-binding signal during the allosteric activation of PKA. Altogether, this study illustrates how each structurally conserved CNB domain has evolved to carry out unique but mutually coordinated regulatory tasks in a macromolecular assembly. Our work reveals new operating principles for ligand-directed protein allostery mediated by widely conserved signaling modules.

## Results

**Optical tweezers assay to extract folding energy landscapes**. To study CNB domain communication mechanisms triggered by cAMP, we perturbed the free energy landscape of the PKA regulatory subunit with optical tweezers (Fig. 1a, b, left). We attached DNA handles via thiol chemistry to two cysteines engineered at specific positions in the protein (see Methods)[13,14]. The handle position determines the direction and region of the protein subjected to the force applied through the optical tweezers (i.e., pulling geometry)[15–17]. We generated three PKA regulatory subunit constructs with unique pulling geometries to probe cAMP binding coupled to inter-domain interactions (Fig. 1b, right). In type-I constructs, force is applied to the isolated CNB domains to study the effect of cAMP binding on the free energy landscape of each domain. In type-II constructs, force is applied selectively to one CNB domain in the presence of the neighboring

one. This pulling geometry allows us to directly assess how cAMP binding induces inter-domain interactions, a strategy that would otherwise be inaccessible with bulk methods or single-molecule fluorescence techniques. In type-III constructs, force is applied across both CNB domains simultaneously, allowing non-contiguous regions of the protein to respond to force, thereby probing long-range allosteric interactions, either in the presence or absence of cAMP.

We separately tethered each type of protein construct between two polystyrene beads in the optical tweezers (Fig. 1b, left and Methods). By gradually increasing and decreasing the tension across a single protein ("force-ramp" pulling), we observed one or more rips in the resulting force-extension curves that correspond to unfolding and refolding events, respectively (Fig. 1c). In the apo state, the isolated CNB domains unfold at a similar average force, $F_{avg} \sim 7–9$ pN, and with similar unfolding kinetic parameters the lifetime of the folded state extrapolated to zero force, $\tau_{0,F}$, is $1.1–1.6 \cdot 10^3$ s and the distance to the transition state, $\Delta x^{\ddagger}_{F \to U}$, 4–5 nm (Fig. 1d and Supplementary Table 1)[18,19]. Analysis of refolding transitions show small differences, wherein the isolated CNB-A domain has a shorter unfolded-state lifetime, $\tau_{0,U}$, and a longer $\Delta x^{\ddagger}_{U \to F}$ compared with the isolated CNB-B domain (Supplementary Table 1). The selective (type-II constructs) or simultaneous (type-III construct) mechanical manipulation of the CNB domains in the apo state showed indistinguishable unfolding and refolding kinetic parameters compared with their isolated counterparts (Fig. 1e and Supplementary Figs. 1–2), indicating that inter-domain interactions within the PKA regulatory subunit are negligible in the absence of cAMP[20].

**Asymmetric domain stabilization effects triggered by cAMP**. In contrast to the results obtained in the apo state, the presence of cAMP revealed important differences between the two CNB domains. The unfolding force of the isolated CNB-B domain increases to $F_{avg} = 12.0 \pm 1.0$ pN (Fig. 2a, b, $N = 648$), resulting in a $\sim 30$-fold increase of $\tau_{0,F}$ (Fig. 2c and Supplementary Table 2). For the isolated CNB-A domain, $F_{avg} = 17.4 \pm 2.0$ pN (Fig. 2d, e, $N = 785$) and $\tau_{0,F}$ increases by a factor of $\sim 7$ (Fig. 2f). The kinetic stabilization conferred by cAMP is also observed during the refolding reaction; both CNB domains had a $\sim 4$-fold decrease in $\tau_{0,U}$ (Supplementary Fig. 1).

Having characterized each isolated CNB domain, we studied inter-domain interactions triggered by cAMP using type-II constructs. We find that both CNB domains were stabilized by the presence of their counterpart when bound to the cyclic nucleotide (Fig. 2a–f). Interestingly, the magnitude of stabilization was asymmetric (Supplementary Table 2): The CNB-A domain stabilizes the CNB-B domain by an additional $\sim 8$ pN, resulting in $F_{avg} = 19.7 \pm 1.6$ pN ($N = 1518$) and a fourfold increase in $\tau_{0,F}$. The presence of the CNB-B domain induces a mechanical stabilization to the CNB-A domain of $\sim 3$ pN, resulting in $F_{avg} = 20.3 \pm 1.4$ pN ($N = 1152$) and a 160-fold increase in $\tau_0$. In the refolding reaction, the presence of the neighboring domain is highly asymmetric, decreasing $\tau_{0,U}$ by 150-fold and 10-fold to the CNB-B and CNB-A domains, respectively (Supplementary Fig. 1).

Having obtained the lifetimes of the folded ($\tau_{0,F}$) and unfolded states ($\tau_{0,U}$) at zero force for type-I and type-II constructs (Supplementary Tables 1 and 2), we dissected the contribution of cAMP binding and inter-domain contacts to the equilibrium free energy and folding energy landscape of each CNB domain (Supplementary Methods). cAMP binding stabilizes the CNB-B domain from 7.6 to 10.4 kcal mol$^{-1}$ and the presence of the neighboring cAMP-bound CNB-A domain provides another

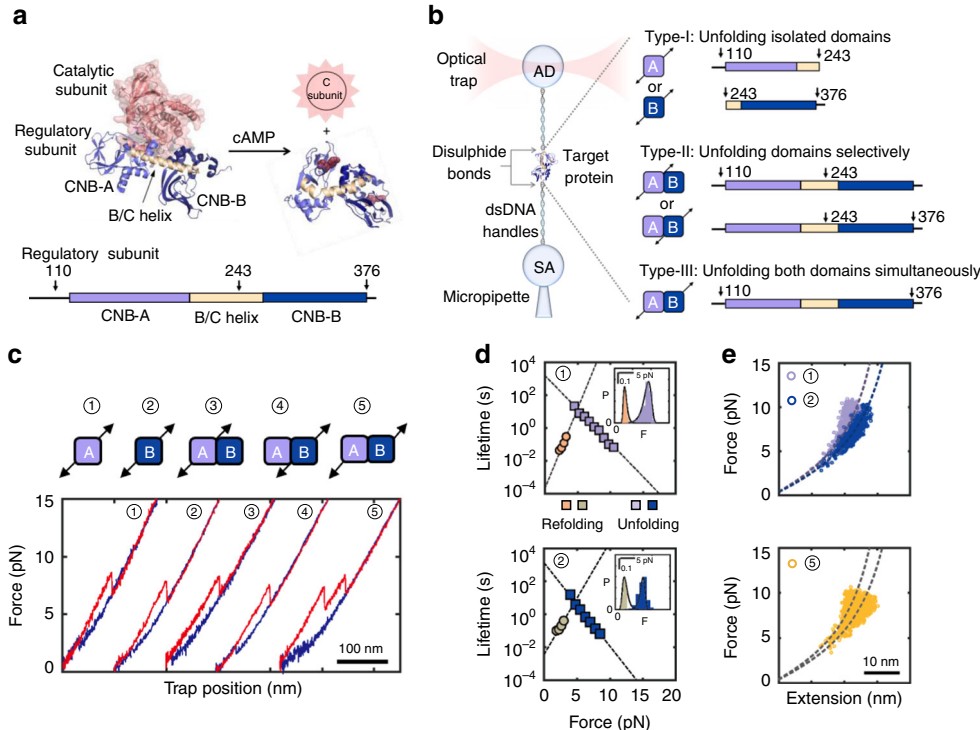

**Fig. 1** Experimental design to study allosteric activation in PKA with optical tweezers. **a** Structure of the inactive PKA holoenzyme (left)[10] and active cAMP-bound regulatory subunit (right)[11]. The arrows on regulatory subunit domain organization indicate the residue positions for DNA handle attachment (bottom, arrows). Source data are provided as a Source Data file. **b** Schematic representation of optical tweezers assay (left) and protein constructs used in this study (right). **c** Force-extension curves for all pulling geometries in the apo state (unfolding in red; refolding in blue). Numbers match the pulling geometry with the unfolding and refolding trajectories. **d** Force-dependent folded-state lifetimes and unfolding force probability distribution (inset) in the apo state. Black lines in insets are the unfolding force distribution reconstructed from force-dependent lifetimes. **e** Worm-like chain (WLC) analysis of changes in extension vs. force for the isolated CNB domains (top) and for the first and second unfolding rips from the type-III construct (bottom). Dashed lines are the WLC curves for the CNB-A (purple) and CNB-B domains (blue). Numbering in **d** and **e** is the same as in **c**

3.2 kcal mol$^{-1}$. We estimate that cAMP binding stabilizes the CNB-A domain from 9.4 to 11.5 kcal mol$^{-1}$ and inter-domain interactions confer an additional 4.4 kcal mol$^{-1}$ (Supplementary Tables 1 and 2). The stabilities estimated in this study for the isolated, cAMP-bound CNB domain are in agreement with previous bulk studies[21–23].

Altogether, these results illustrate that the minor structural differences between the two CNB domains (root mean square deviation (RMSD) = 1.2 Å between Cα atoms, Fig. 2g) do not reflect the important differences in the folding energy landscape response to cAMP binding and inter-domain contacts (Fig. 2h). In fact, our results show that cAMP binding induces specific but coordinated effects, wherein the CNB-B domain stabilizes the folded state of the CNB-A domain and the CNB-A domain destabilizes the unfolded state of the CNB-B domain. Therefore, the cAMP-dependent communications between the CNB domains is bidirectional and asymmetric, highlighting a unique role for each domain in the activation mechanism of PKA.

**Identification of a cAMP-responsive dynamic switch**. We hypothesized that changes in contour length upon unfolding ($\Delta L_c$) might also reveal important differences in the native folded structures of the CNB domains upon binding cAMP. Although the mechanical unfolding of the CNB-B domain in all three types of constructs had a $\Delta L_c$ of ~50 nm, corresponding to a fully folded domain (Supplementary Table 2), the CNB-A domain displayed a more complex behavior. The isolated CNB-A domain in the apo state had the expected $\Delta L_c$ of 45 nm based on the crystal structure[10]. However, the value of $\Delta L_c$ decreased to 30 ±

3 nm in the presence of cAMP, indicating that a region of the domain was destabilized upon ligand binding (Supplementary Table 2).

We sought to identify which region or secondary structures of the CNB-A domain become unstable upon cAMP binding. The structure of the CNB-A domain is composed of a β-sandwich fold that forms the cAMP-binding pocket and three N-terminal α-helices termed N3A motif[10,11]. The N3A motif contains ~30 amino acids, which matches the amount of polypeptide that became unstable after cAMP binding. To test whether the N3A motif is destabilized by cAMP binding, we used two distinct type-III constructs, one with DNA handles attached at residue positions flanking both CNB domains entirely (S110C/S376C) and another construct with handles flanking both CNB domains, except the N3A motif (D149C/S376C). The two constructs displayed two major unfolding rips corresponding to the CNB domains, but only the unfolding trajectory of the S110C/S376C construct revealed a small, reversible transition at ~11 pN with a $\Delta L_c$ of 13 nm (Fig. 3a). The lack of such small transition in the D149C/S376C construct, which does not directly probe the N3A motif, provides evidence that the secondary structures in the CNB-A domain that become unstable upon cAMP binding correspond to the N3A motif (Supplementary Fig. 3).

Standard molecular dynamic (MD) simulations starting from the X-ray structure reveals that the unbound state (apo) during the relaxation undergone to a conformational rearrangement involving the loss of interactions between Trp260 and cAMP docked into CNB-A (Supplementary Fig. 4a, b). The cAMP-bound state adopted a closer, more compact shape than the apo state (Supplementary Fig. 4b). The compact state results from a

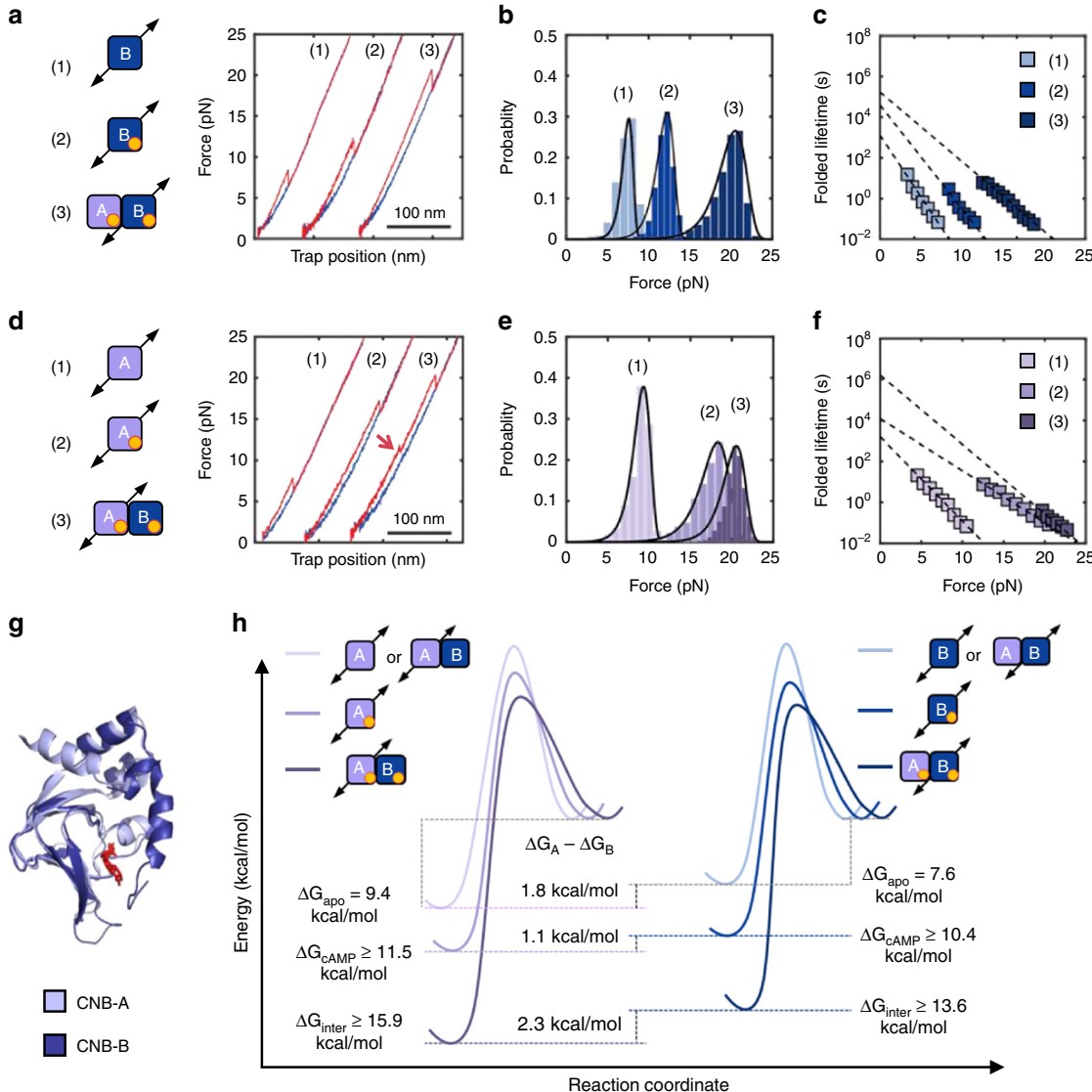

**Fig. 2** Selective allosteric effects initiated by cAMP binding. Force-extension curves (**a**, **d**), unfolding force probability distributions (**b**, **e**), and force-dependent folded-state lifetimes (**c**, **f**) for the CNB-B (top) and CNB-A (bottom) domains. Numbering corresponds to the isolated CNB domains in the apo (1) or cAMP-bound states (2), and selective unfolding of the CNB domains bound to cAMP (3). The red arrow in **d** indicates the unfolding of the N3A motif. Source data for **a** through **d** are provided as a Source Data file. **g** Structural alignment of the CNB-A (light purple) and CNB-B (dark blue) domains bound to cAMP (red). **h** Energy landscape and free energy of the CNB domains due to cAMP binding and inter-domain contacts. The height of the energy barriers reflects the folded and unfolded-state lifetimes of the CNB domains in the different states. The energy landscapes have been normalized to the unfolded state. (Supplementary Tables 1 and 2)

more favorable interaction energy between the CNB domains, being the inter-domain interaction of the cAMP-bound state significantly more stable than that of the apo state. A set of four SMD simulations starting from different conformations selected from the MD relaxation of the X-ray structure (Supplementary Fig. 4a) corroborate our experimental observations (see also Supplementary Movies 1 and 2). Cluster analysis performed over the SMD trajectories show that the N3A motif unravels first, whereas the rest of the CNB domains remained stably folded in their original cAMP-bound conformation (Fig. 3b, right). In the absence of cAMP, several inter-domain interactions were lost (Supplementary Fig. 4c), resulting in the detachment of the two CNB domains before any secondary structure unfolds, including the N3A motif (Fig. 3b, left).

**N3A motif folding requires complete inter-domain contacts.** In contrast to the results obtained with the isolated cAMP-bound

CNB-A domain, the type-III S110C/S376C construct show that the N3A motif is properly folded in the context of the entire regularly subunit. Moreover, a close inspection of trajectories obtained with the selective manipulation of the cAMP-bound CNB-A domain (type-II construct) revealed a two-step unfolding process instead of a single rip (Fig. 2d, red arrow). The additional rip had a $\Delta L_c$ of ~13 nm, similar to that of the N3A motif. These observations indicate that the CNB-B domain enables the refolding of the N3A motif in the presence of cAMP.

As our optical tweezers assay permitted to control the sequence of events in the unfolding reaction, we used the type-III S110C/S376C construct to determine whether the N3A motif can refold while the CNB-B domain remains in the unfolded state (Fig. 4a). In this experiment, we applied a force up to 15 pN, to unfold both the N3A motif and the CNB-B domain, but not the CNB-A domain. The unique, reversible transition of the N3A motif indicates that this motif hops between the folded and unfolded

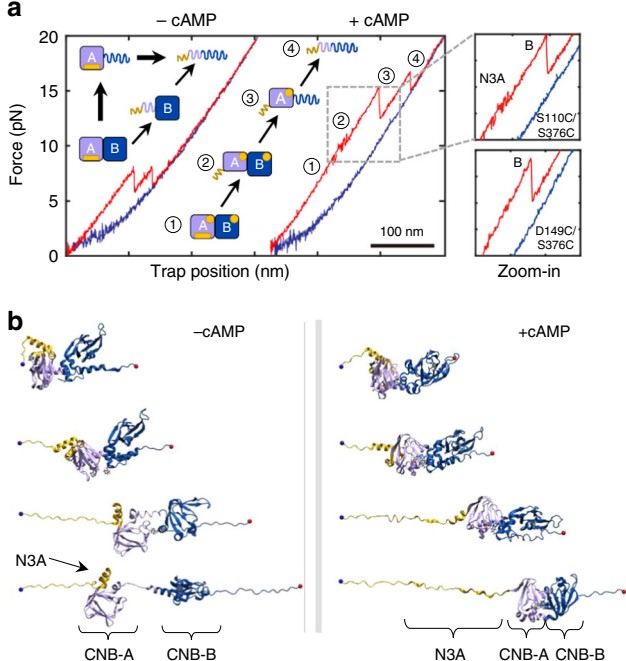

**Fig. 3** Dual unfolding pathways of PKA regulatory subunit depending on cAMP occupancy. **a** Force-extension curve for the cAMP-free (left) and cAMP-bound (right) regulatory subunit (type-III construct). The cartoon represents the structural transitions occurring during the unfolding trajectory in apo- and cAMP-bound states. Zoomed-in are the unfolding trajectories of type-III constructs S110C/S376C and D149C/S376C bound to cAMP. A detailed analysis of the forces for each unfolding transition refer to Supplementary Information and Supplementary Table 4. **b** Representative structures from cluster analysis along the SMD trajectories of the cAMP-bound (left) and apo (right) regulatory subunit. Yellow: N3A motif; Purple: CNB-A; Dark blue: CNB-B domain

states in the force range of 10–12 pN. The force was then decreased to 5 pN, to maintain the CNB-B domain in the unfolded state (refolding transitions begin at forces <2 pN). After ten or more pulling and relaxation cycles between 5 and 15 pN, we did not observe any small, reversible transitions at ~11 pN, which would have corresponded to a folded N3A motif, while the CNB-B domain remained unfolded. Thus, we find that the CNB-B domain is strictly required for the N3A motif to refold. This result is in agreement with the structure of the cAMP-bound regulatory subunit that shows the N3A motif docks into a cleft formed between the CNB domains[11], establishing several surface contacts not only with the CNB-A domain and the B/C helix[24] but also with the CNB-B domain (Fig. 4b).

**Inter-domain Ccontacts begin in partial cAMP-bound states.** As cAMP binds to PKA in a sequential manner[10], thereby populating intermediate cAMP-bound species, we investigated the coupling between the folding status of the N3A motif and inter-domain interactions in conditions where only one CNB domain is bound to cAMP. To obtain force-extension curves of intermediate cAMP-bound states, we used the type-III construct S110C/S376C and titrated cAMP between 1 and 150 nM. Based on their unique unfolding forces and $\Delta L_c$, we were able to identify distinct states where only one CNB domain is bound to cAMP among all possible liganded states: apo, only CNB-A domain, only CNB-B domain bound, or both (Fig. 5a and Supplementary Methods). We find that the CNB-B domain bound to cAMP increases both $F_{avg}$ by ~1 pN (KS test, $p \cong 0$) and $\tau_{0,F}$ by twofold to the cAMP-free CNB-A domain (Fig. 5b, top). The CNB-A

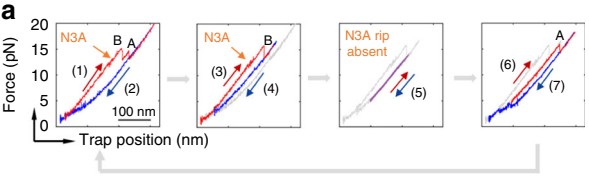

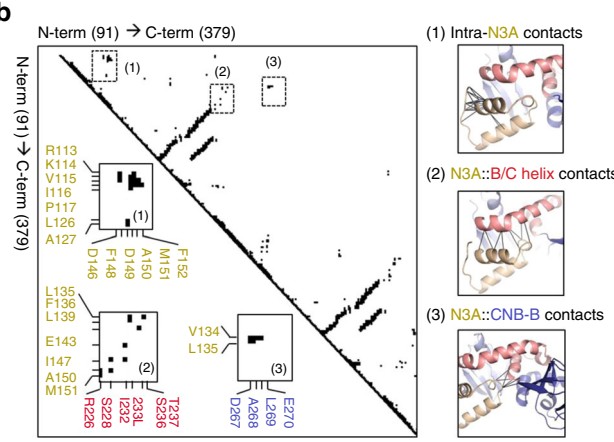

**Fig. 4** The CNB-B domain bound to cAMP is required for the N3A motif to fold. **a** (1) The regulatory subunit unfolds (red) and (2) refolds (blue), revealing the first reversible transition corresponding to the N3A motif (orange arrow). (3) In the following cycle, the regulatory subunit was stretched until the N3A motif and the CNB-B domain unfold, whereas the CNB-A domain remains folded. (4) The force was decreased to 4 pN, a force that does not allow the CNB-B domain to refold. (5) The force was then oscillated between 6 and 15 pN for several cycles (~20) to test whether the N3A motif was able to refold, while the CNB-B domains remain unfolded. (6) The force was increased to 20 pN to unfold the CNB-A domain. (7) The force was decreased to 1 pN, allowing the complete protein to refold and begin another set of experiments. The trajectories in gray represent the unfolding pathways from the immediately previous cycle, thereby serving as reference on the progression of experiment. **b** Pairwise contact map comparing the interaction established by the N3A motif in the regulatory subunit of PKA (left). The contacts established by the N3A motif were obtained using a 8 Å cutoff (right): (1) contacts established by residues within the N3A motif; (2) contacts between the N3A motif and the B/C helix; and (3) contacts between the N3A motif and the CNB-B domain. Cartoons rendering the three sets of contacts are shown next to the contact map. Residues in the N3A motif are colored in yellow, in the B/C helix in red, and in the CNB-B domain in blue

domain bound to cAMP induces a larger stabilization to the cAMP-free CNB-B domain, increasing $F_{avg}$ by ~3 pN and $\tau_{0,F}$ by threefold (Fig. 5b, bottom). Previous computational studies have identified asymmetric interactions in intermediate-liganded states, where the cAMP-bound CNB-A domain maintains inter-domain contacts similar to those for the doubly bound form[25]. Our results support these simulations, showing that cAMP binding to one CNB domain is sufficient to initiate stabilizing inter-domain interactions with the neighboring apo CNB domain; however, compared with the fully cAMP-bound state, these interactions are partial in magnitude (Supplementary Table 3). Moreover, we find that these partial inter-domain interactions are insufficient to drive the folding of the N3A motif between the two CNB domains, i.e., analysis of $\Delta L_c$ using the Worm-like chain model shows that the cAMP-bound CNB-A domain interacting with the cAMP-free CNB-B domain does not have a folded N3A motif (Fig. 5c, top). A similar analysis revealed that cAMP binding to the CNB-B domain does not elicit unfolding of the N3A motif in the cAMP-free CNB-A domain (Fig. 5c, bottom).

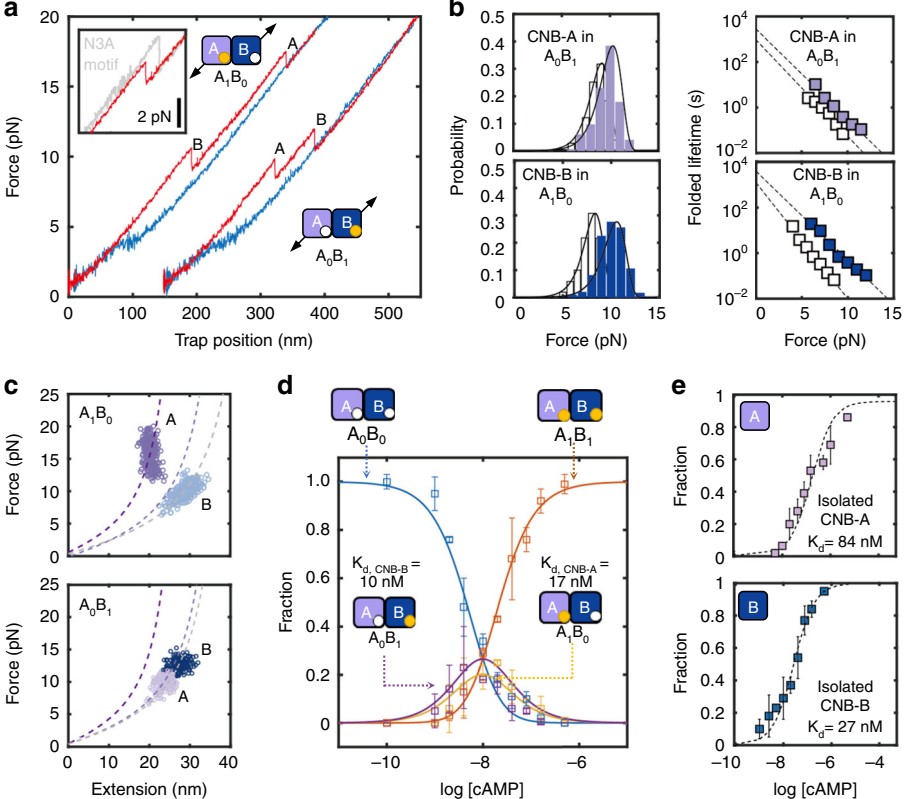

**Fig. 5** Stepwise stabilization between CNB domains in partial cAMP-bound states. **a** Representative force-extension curves of two intermediate-liganded states ($A_1B_0$ and $A_0B_1$) for the type-III regulatory subunit S110C/S376C. Zoomed-in are the unfolding trajectories of $A_1B_0$ compared with fully bound state (i.e., $A_1B_1$), showing the lack of N3A motif hopping in $A_1B_0$. **b** Unfolding force probability distribution and force-dependent folded-state lifetimes for intermediate-liganded states: CNB-A domain in $A_0B_1$ (top) and CNB-B domain in $A_1B_0$ (bottom). The corresponding isolated domains in the apo state (white bars and symbols) are shown for comparison. Solid lines are the unfolding force distribution reconstructed from force-dependent lifetimes. **c** WLC analysis of changes in extension upon unfolding vs. force in $A_1B_0$ and $A_0B_1$ for the CNB-A and CNB-B domains (dashed lines). Expected WLC curves are shown for CNB-A domain without the N3A motif (dark purple) and with the N3A motif (light purple). **d** cAMP titration plot showing the fraction of apo ($A_0B_0$), intermediate ($A_1B_0$ or $A_0B_1$), and fully bound ($A_1B_1$) species. Lines correspond to the global fit to the equations for each population species (Supplementary Methods). Error bars are the weighted SD of different single molecules. **e** Fractional titration plot of isolated CNB-A (top) and CNB-B (bottom) domains (type-I constructs). The error bar corresponds to the SD of five to ten different molecules

These results strongly support our previous observations showing that unfolding of the N3A motif is solely coupled to cAMP binding to the CNB-A domain, and that the following refolding step of the N3A motif requires the presence of the cAMP-bound CNB-B domain (Fig. 4a).

We also find that partial inter-domain interactions initiated by on-pathway intermediate cAMP-bound states have important functional consequences in terms of cAMP-binding affinities and cooperativity. By directly counting unfolding trajectories corresponding to the apo, intermediate- and fully bound species as a function of cAMP concentration (Supplementary Fig. 5), we built a single-molecule titration curve, globally fitted the equations for each population species, and determine the microscopic binding constants and cooperativity parameter (Fig. 5d and Supplementary Methods). For the first cAMP molecule, the CNB-B domain has a dissociation constant $K_{d,CNB-B}$ of $10 \pm 1$ nM and the CNB-A domain has a $K_{d,CNB-A} = 17 \pm 1$ nM. The $K_d$ for the second cAMP molecule for either CNB domain is approximately threefold lower, indicating positive binding cooperativity. Importantly, the $K_d$ values of the CNB domains are three and five times lower than those corresponding to the isolated domains, respectively (Supplementary Methods), indicating that as part of the regulatory subunit the CNB domains bind cAMP more tightly (Fig. 5e). The single-molecule titrations and the extracted microscopic binding affinities for the isolated CNB domains are

consistent with previous reports on $EC_{50}$ for cAMP[26]. Our single-molecule studies, however, provide in addition direct access to individual binding events to each CNB domain as part of the regulatory subunit and the cooperativity involved in the cAMP-binding reaction.

**N3A motif dynamic motions is critical for PKA activation**. Our results portray the N3A motif as a ligand-responsive molecular switch that toggles between different conformations depending on cAMP occupancy and specific domain contacts. This unique character led us to hypothesize that the N3A motif is a critical structural element that mediates cAMP-dependent cooperative interactions between the CNB domains. We tested this hypothesis by placing the mutation R241A in the B/C helix that connects both CNB domains. In the wild-type structure bound to cAMP, R241 interacts with D267 in the CNB-B domain and E200 in the CNB-A domain, thereby bringing the two CNB domains into close proximity for the N3A motif to dock (Fig. 6a, left)[11,27]. In the absence of cAMP, unfolding trajectories of R241A using a type-III construct (S110C/S376C) show indistinguishable unfolding parameters compared with wild-type (Supplementary Fig. 6 and Supplementary Table 4). In the presence of cAMP, the trajectories of R241A revealed an unfolding pathway that looked similar to that of wild type, but with some important quantitative

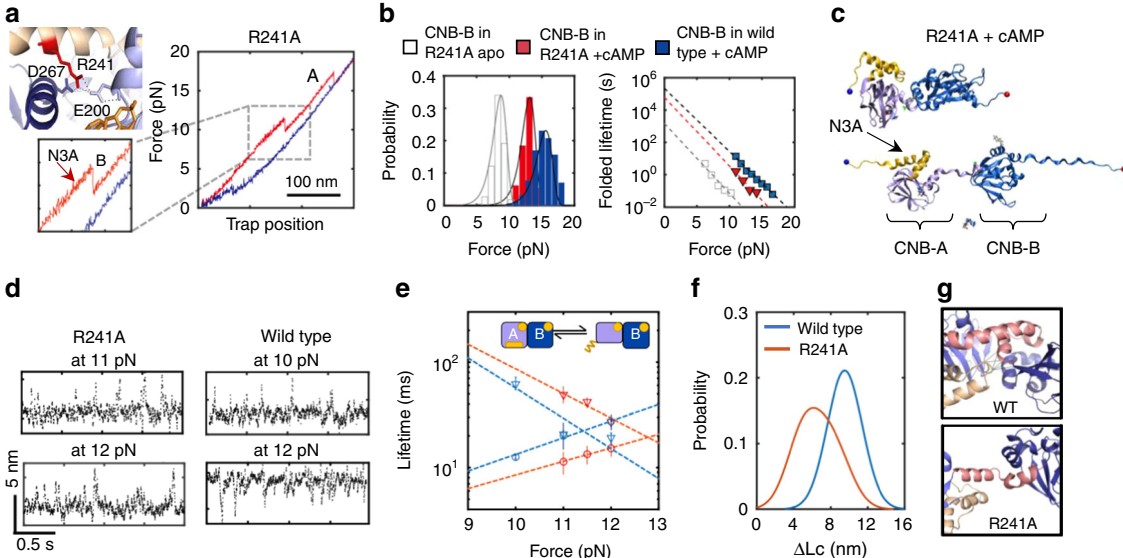

**Fig. 6** Perturbation of allosteric networks in PKA by mutation R241A. **a** Residue R241 interacts with both CNB domains through E200 and D267 (PDB 1RGS). Force-extension curve for R241A bound to cAMP (type-III construct S110C/S376C). Zoomed-in is the unfolding rip corresponding to the N3A motif. **b** Unfolding probability distributions and force-dependent folded-state lifetimes for the CNB-B domain in R241A apo (gray) or bound to cAMP (red). For reference, the wild-type data were included (blue). Solid lines are the unfolding force distribution reconstructed from force-dependent lifetimes. **c** SMD simulation snapshot of the cAMP-bound R241A protein. **d** Representative force-clamp trajectories of the N3A motif in R241A and wild type. Source data are provided as a Source Data file. **e** Force-dependent lifetimes of the N3A motif in the folded (triangles) and unfolded (circles) states for R241A (red) and wild type (blue). Error bars are the SD of different single molecules. **f** Distribution of the change in contour length ($\Delta L_c$) of the N3A motif for wild type (blue) and R241A (red). **g** The mutation R241A (bottom) hinders interactions established with the CNB-B domain seen in wild type (top). Source data for **d** and **g** are provided as a Source Data file

differences (Fig. 6a, right, and Supplementary Fig. 6). Specifically, the average unfolding force for the CNB-B domain in the mutant protein was ~2.5 pN lower compared with wild type, which results in a threefold reduction of $\tau_{0,F}$ (Fig. 6b). These values are indistinguishable from those obtained with the isolated CNB-B domain bound to cAMP (Supplementary Table 4), indicating that R241A largely eliminates inter-domain interactions initiated by the cyclic nucleotide. SMD trajectories also show that the first event in the unfolding pathway of R241A is the detachment of the cAMP-bound CNB domains, instead of the unraveling of the N3A motif (Fig. 6c, Supplementary Fig. 4d, and Supplementary Movie 3).

To further dissect the role of folding dynamics and conformation of the N3A motif in inter-domain interactions, we conducted "force-clamp" experiments, wherein the protein was held at varying constant forces between 10 and 12 pN, and changes in extension due to unfolding and refolding of the N3A motif were monitored as a function of time (Fig. 6d). Analysis of these trajectories using a two-state Bayesian Hidden Markov model (BHMM)[28,29] revealed that the folded-state lifetime was approximately two times longer and the unfolded-state lifetime was approximately twofold shorter for R241A, indicating that the N3A motif in the mutant protein is less dynamic than in wild type (Fig. 6e). In addition, the accompanying $\Delta L_c$ between the folded and unfolded states for R241A was $6.5 \pm 1.1$ nm and for wild type was $9.5 \pm 0.5$ nm. The difference in folding dynamics and $\Delta L_c$ indicates that the N3A motif in R241A is stably folded. However, as the mutation eliminates cAMP-dependent inter-domain interactions (Fig. 6b), it is likely that the folded N3A motif is not docked between the two CNB domains (Fig. 6g). Previous studies have shown that to activate PKA, the R241A mutant requires 20-fold more cAMP compared with wild type (activation constant or $K_{a,WT} = 23$ nM, $K_{a,R241A} = 543$ nM)[30]. Although the $K_a$ for the mutant is significantly larger, its cAMP binding affinity is comparable to wild type. In silico studies

proposed that the difference between intrinsic cAMP affinities and activation constants originate from mutational effects over the conformation of the protein. Our findings here show that the R241A mutant imparts cAMP-dependent functional deficiencies due to a disruption of the conformational dynamics of the N3A motif and its ability to serve as an efficient cAMP-responsive molecular switch, thereby impeding the PKA regulatory subunit to attain its final cAMP-bound conformation.

**Dynamic switching of the N3A motif is selective to cAMP.** Previous mutational studies guided by the high-resolution structures of the regulatory subunits of PKA and the homolog cGMP-dependent protein kinase have allowed the identification of residues important for CNB affinity and selectivity.[31–32] These studies have shown that cGMP binding to PKA is weaker than for cAMP at the level of individual CNB domains ($K_{d,CNB-A(cAMP)} = 4.2$ nM, $K_{d,CNB-A(cGMP)} = 820$ nM, $K_{d,CNB-B(cAMP)} = 2.8$ nM, and $K_{d,CNB-B(cGMP)} = 230$ nM)[33,34]. Moreover, the activation constant of PKA by cGMP is ~140-fold higher than that of cAMP ($K_{a,cAMP} = 53$ nM and $K_{a,cGMP} = 7400$ nM)[32]. Less understood, however, is the dependence of the conformational changes that the PKA regulatory subunit experiences upon binding cAMP vs. cGMP. Therefore, we investigated the contribution of the N3A motif dynamic switch mechanism towards cyclic nucleotide selectivity by mechanically manipulating the CNB domains individually (type-I constructs) or simultaneously (type-III construct) in the presence of cGMP. Both CNB domains bound to cGMP show unfolding parameters ($F_{avg}$, $\tau_{0,F}$, and $\Delta x^{\ddagger}_{F \to U}$) that lie in between the values obtained with and without cAMP, indicating partial intra- and inter-domain stabilization effects (Fig. 7a, b and Supplementary Table 5). Interestingly, the isolated cGMP-bound CNB-A domain had a greater $\Delta L_c$ than its cAMP-bound counterpart (37 nm and 30 nm, respectively), indicating that the N3A motif is not negatively coupled to cGMP, but

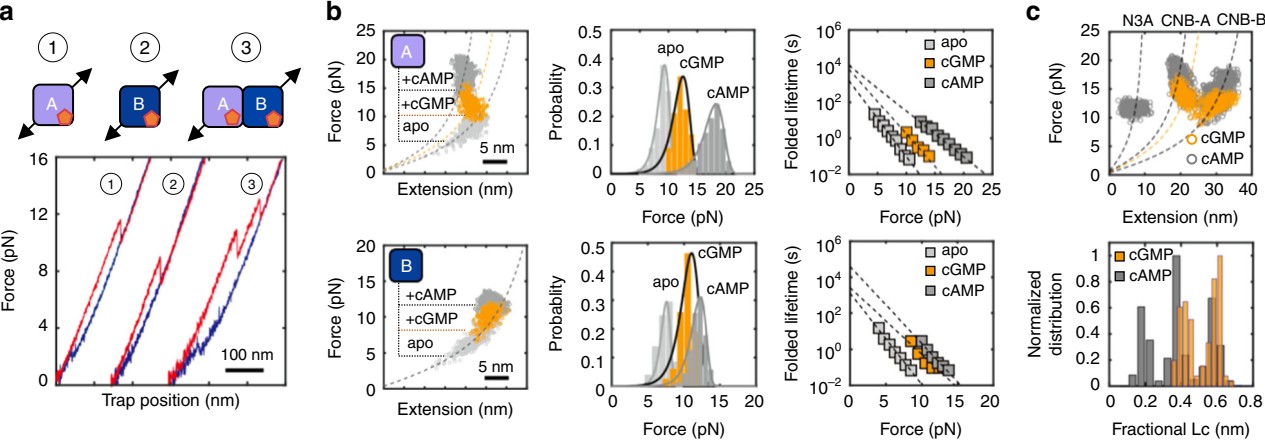

**Fig. 7** Perturbation of allosteric networks in PKA by cGMP. **a** Force-extension curves of cGMP-bound protein constructs. **b** WLC analysis of changes in extension upon unfolding vs. force (left), unfolding force probability distributions (center) and force-dependent folded-state lifetimes (right) for the isolated CNB-A (top) and CNB-B (bottom) domains in the apo (light gray), cGMP-bound (orange), and cAMP-bound (dark gray) states. Solid lines in center panels are the unfolding force distribution reconstructed from force-dependent lifetimes. **c** WLC analysis of changes in extension upon unfolding vs. force (top) and fractional contour length (bottom) of the regulatory subunit bound to cGMP (orange) and cAMP (gray). Dashed lines are the WLC curves for the N3A motif and the two CNB domains with cAMP (gray) and cGMP (orange). Source data are provided as a Source Data file

instead unfolds as a single cooperative unit together with the rest of the domain. In agreement with this interpretation, the force-extension curves using a type-III construct with cGMP do not show the small, reversible transition characteristic of the N3A motif (Fig. 7a). Rather, the trajectory revealed two unfolding rips with $\Delta L_c$ values that reflect the mechanical denaturation of the full-length protein (Fig. 7c and Supplementary Fig. 7). These results provide direct experimental evidence that nucleotide selectivity not only involves previously described defects in binding affinity[33] but also an attenuation of inter-domain interactions and decoupling of CNB from the conformational switching of the N3A motif.

## Discussion

The uncovered intra- and inter-domain communication network that is triggered by cAMP binding cannot be easily inferred from crystal structures[9,10,12,35,36]. The network of interactions in PKA involves bidirectional communication that is asymmetric in magnitude and includes both positive (stabilizing) and negative (destabilizing) coupling interactions that are fine-tuned to attain the final cAMP-bound conformation.

Positive coupling arises from cAMP binding, stabilizing interfacial interactions between CNB domains, and the conformational reorganization of the N3A motif between the two cAMP-bound CNB domains (Fig. 4b). By mechanically unfolding and refolding each CNB domain as an isolated structure or selectively in the regulatory subunit (type-I and type-II constructs, respectively), we dissected the contribution of cAMP binding and inter-domain interaction to the stability of each CNB domain (Fig. 2h). The thermodynamic stability of the isolated CNB domains bound to cAMP is comparable to values obtained in previous nuclear magnetic resonance and urea denaturation studies[21,22]. In addition, our approach of selectively manipulating an individual CNB domain in the presence of the neighboring one allowed us to directly quantify interfacial stabilization effects due to cAMP binding. By using a type-III construct, we further determined that the N3A motif contributes 5.5 kcal mol$^{-1}$ to the stability of the protein, allowing us to propose a folding energy landscape of the regulatory subunit when force is applied to both CNB domains simultaneously (Fig. 8a). Negative coupling interactions arise from cAMP binding to the CNB-A domain and

results in the destabilization of the N3A motif. Such destabilizing effect may be important to weaken extensive surface contacts between the N3A motif and the catalytic subunit (Supplementary Methods), thereby facilitating the dissociation of the PKA complex.

Based on our studies, a critical element in the allosteric activation mechanism of PKA involves the dynamic switching of the N3A motif, which we show is important to stabilize the cAMP-bound conformation of the regulatory subunit. Previous structural and computational studies have identified substantial rearrangements and motions of the N3A motif during the allosteric signals triggered by cAMP binding[10,21]. In those studies, it is proposed that the binding of the first cAMP molecule to the CNB-B domain induces a conformational change in the B/C helix, breaking critical inter-subunit hydrogen bonds and van der Waals interactions between the B/C helix and the catalytic subunit[37]. The conformational change of the B/C helix may propagate to the N3A motif in the CNB-A domain, which provides a mechanism of communication between the CNB domains in the PKA holoenzyme[10,21,38]. The N3A motif in the CNB-A domain is a major contributor to the interaction surface established between the catalytic and regulatory subunit in the inactive PKA holoenzyme. Moreover, the N3A motif bound to the catalytic subunit makes contacts with residues in the CNB-A domain that are important for cAMP binding (i.e., N133 in the N3A motif and E200 in the CNB-A domain). Therefore, the dissociation from the catalytic subunit and the conformational rearrangement of the N3A motif to allow interactions with cAMP must be critical steps in the activation mechanism of PKA. Thus, our studies show that the N3A motif behaves like a dynamic switch, serving as a signaling hub that amplifies the cAMP-binding signal during the allosteric activation of PKA. By revealing the dynamic motions and stability changes that the N3A motif experiences as cAMP binds to the CNB-A domain and triggers inter-domain interactions with the CNB-B domain, our studies provide direct measurements on this conformational switch. Given the remarkable structural similarity between different regulatory subunit isoforms bound to the catalytic subunit (RMSD ~ 0.5–0.6 Å between RIα–RIIα or RIα–RIIβ)[10,12,39], we proposed that the N3A motif plays a similar dynamic-switching role to activate other PKA isoforms. Moreover, as the N3A motif is found in many other cAMP- and cGMP-binding proteins[40], the uncovered

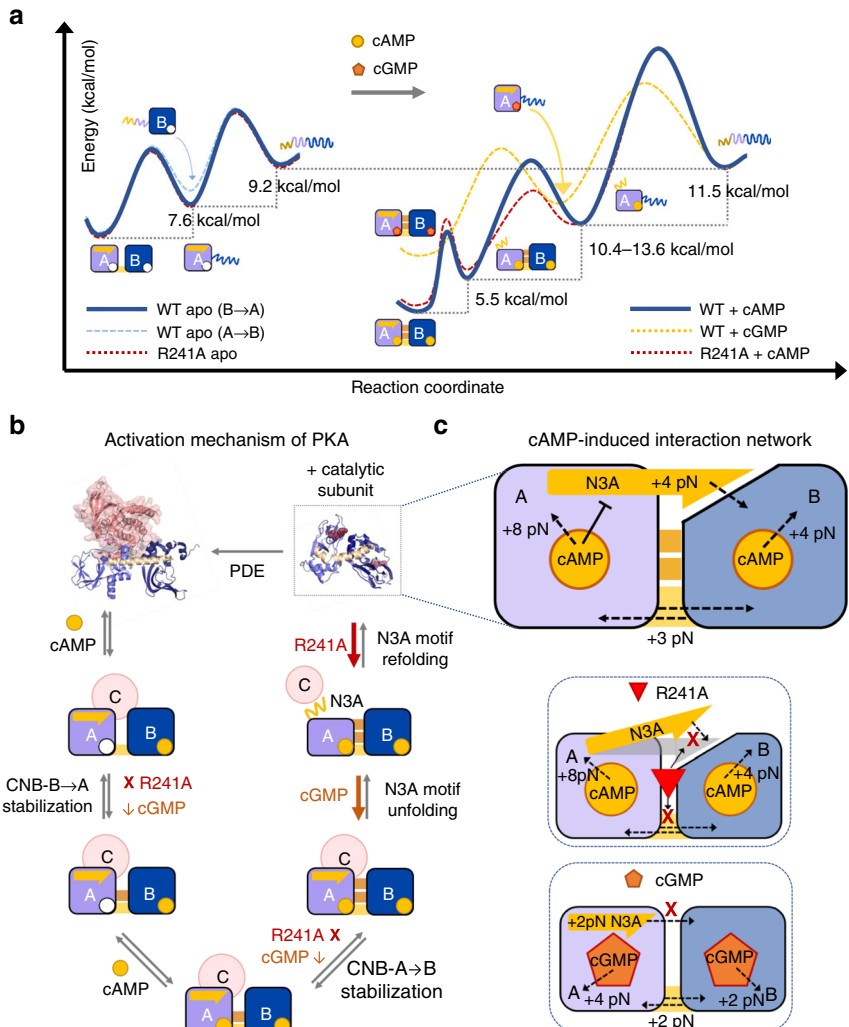

**Fig. 8** Activation of PKA through selective stabilization of CNB domains. **a** Unfolding energy landscape of the regulatory subunit in the apo state (left) and bound to cyclic nucleotide (right). The unfolding pathway in the apo state follows a CNB-B-to-CNB-A order in 80% of events (solid blue line). In the other 20%, the apo CNB-A domain unfolds first (dashed light blue line). The apo R241A mutant (dashed red line) has an indistinguishable energy landscape compared with apo wild type. In the presence of cAMP, the unfolding order in wild type is as follows: the N3A motif, the CNB-B domain, and the CNB-A domain. The cAMP-bound R241A mutant follows a similar unfolding order, but the CNB-B domain unfolds with a lower energy barrier and has lower stability due to the lack of stabilizing inter-domain interactions. The cGMP-bound wild type follows an unfolding pathway that is quantitatively and qualitatively different to that of the cAMP state (see main text for details). The height of the energy barriers reflects the folded and unfolded-state lifetimes of the CNB domains (Supplementary Tables 2–5). The energy landscapes have been normalized to the unfolded state. **b** Activation mechanism of PKA showing intermediate states and their cAMP-initiated interactions. Steps and interactions disrupted by R241A and cGMP are shown as "X" (complete disruption) or "down-arrow" (decreased in magnitude). **c** Top: interaction network initiated by cAMP binding involves stabilizing (dashed black arrows) and destabilizing (flat black arrow) coupling (top). Effect of R241A (middle) and cGMP (bottom) on interaction network

conformational switching mechanism may be a widespread strategy not only to ensure the completion of the allosteric activation process and provide ligand selectivity.

By integrating all the results from this study, we find an allosteric regulation mechanism that governs the activation of the PKA complex, wherein co-existing positive- and negative-coupling interactions initiated by cAMP binding are coordinated to gradually dissociate the PKA complex (Fig. 8b). These interactions commence at the CNB-B domain, which binds cAMP first and establishes partial inter-domain contacts with the apo CNB-A domain. Partial inter-domain contacts may play multiple roles: promoting the final doubly-bound form (B-form)[11], further facilitating the dissociation of the catalytic subunit and enabling binding of the cyclic nucleotide to the CNB-A domain[10]. After cAMP binds to the CNB-A domain, the conformational switching of the N3A motif is triggered, breaking extensive surface

interactions with the catalytic subunit and stabilizing the final cAMP-bound form[11]. Similar to unzipping "Velcro," this mechanism efficiently peels off the two PKA subunits, where the strong inter-subunit interaction is the result of several smaller interactions that can be broken sequentially; hence, the dissociation of the two subunits does not require the crossing of a large free energy barrier but of many small ones. Within this allosteric regulation network triggered by cAMP binding, we identified the unique routes of domain communication that are disrupted either by R241A or cGMP (Fig. 8c).

The allosteric networks we describe here may be amplified in the PKA hetero-tetramer composed of two regulatory and two catalytic subunits, where the potential cross-talk between PKA subunits is expanded[12,34,40]. Structural studies of the PKA hetero-tetramer formed with RI isoforms showed the N3A motif of one regulatory subunit stacked against the N3A motif of the

neighboring one, forming a helical bundle with several hydrophobic interactions[41,42]. Therefore, it is possible that the dynamic switching response of the N3A motif described here may play additional allosteric regulatory roles by communicating the two regulatory subunits in the PKA hetero-tetramer via quaternary interactions. Consistent with this notion, mutations on the N3A motif (i.e., K121A and Y120A) have been shown to decrease the Hill coefficient of activation of the PKA hetero-tetramer from 1.7 to ~1.0. The mutations S145G and R144S also located in the N3A motif, and that are related to Carney Complex disease have a lower Hill coefficient of 1.4. These mutational studies underscore the functional significance of the N3A motif in the allosteric communication networks and cooperative interactions triggered by cAMP binding[36]. However, communication across regulatory subunits in PKA hetero-tetramers formed with RII isoforms may be different, as the two N3A motifs in the crystal structure do not establish the same quaternary contacts seen in RI isoforms[12].

In a recent study we showed how the CNB domains of the regulatory subunit (RIα) are thermodynamically coupled when bound to the catalytic subunit[20]. The dissection of the folding energy landscape of the CNB domains showed that, when bound to the catalytic subunit, the CNB-B domain controls the stability of the CNB-A domain. This finding provides a thermodynamic foundation by which the CNB-B domain serves as the gatekeeper for cAMP binding to the CNB-A domain[20]. In this study, we use optical tweezers to directly interrogate another aspect of the activation mechanism of PKA, namely the forces that drive the cAMP-mediated activation and conformational changes in the PKA regulatory subunit. Remarkably, the mechanical fingerprints of the regulatory subunit and the underlying folding energy landscapes are unique and depend on whether the protein is in the heterodimeric PKA holoenzyme (i.e., bound to the catalytic subunit) or in the cAMP-bound conformation.

These unique mechanical fingerprints may emerge from the fact that the CNB domains have evolved to behave as molecular switches, changing in conformation upon external stimuli. A case in point is the N3A motif in the CNB-A domain that makes unique contacts depending on whether it is bound to cAMP or to the catalytic subunit. When bound to the catalytic subunit, the N3A motif unfolds together and simultaneously with the β-sandwich of the CNB-A domain, behaving as single cooperative unit[20]. In contrast, the results of this study show that the N3A unfolding behavior is much more complex when bound to cAMP. Thus, the catalytic subunit and cAMP are competing for similar interaction regions in the regulatory subunit, suggesting a "tug-of-war model" in the activation of PKA[21]. Markov state models have suggested the existence of hybrid states, in which the regulatory subunit is bound to the catalytic subunit and partially occupied by cAMP[43]. Future studies will involve characterizing the interaction forces between CNB domains in the PKA hetero-tetramer and in hybrid states with partial cAMP occupancy.

In conclusion, we anticipate similar allosteric regulation mechanisms in other protein kinases with catalytic subunits that require the dissociation of regulatory signaling modules (CNB, SH2, SH3, PH domains, etc.)[44]. The single-molecule approach exploiting optical tweezers in conjunction of molecular dynamics simulations presented here can be extended to map allosteric effects of disease mutations or inhibitor binding in other kinases or multi-domain assemblies[45–49].

## Methods

**Purification of PKA regulatory subunit and isolated CNB domains.** The PRSET plasmid harboring the *Bos taurus* regulatory subunit gene isoform RIα containing residues 91–379 of the full-length sequence was used. To obtain isolated CNB domains, the sequence of the neighboring CNB domain was deleted by site-directed mutagenesis (QuikChange II Agilent). The isolated CNB-A domain

contains residues 91–243. The isolated CNB-B domain contains residues 243–379. The mutations C345A and C360A were introduced in the CNB-B domain to prevent undesired reactions with the thiol-modified double-stranded DNA (dsDNA) handles. We have shown in a previous study that the double mutant C345A/C360A in the CNB-B domain does not alter the solution structure, stability, or the ability to form an inactive complex with the PKA catalytic subunit[20]. In addition, and compared with bulk studies, the CNB-B domain showed a similar binding affinity constant for cAMP either as an isolated domain or as part of the regulatory subunit[50]. To manipulate each individual CNB domain (Type-I constructs), we introduced the mutations S110C/M243C and M243C/S376C for the isolated CNB-A and CNB-B domains, respectively. To manipulate either the CNB-A domain or the CNB-B domain selectively (Type-II constructs), we introduced into the regulatory subunit the mutations S110C/M243C and M243C/S376C, respectively. To manipulate both CNB domains simultaneously (Type-III construct), we introduced into the regulatory subunit the mutations S110C/S376C. All the protein constructs were expressed in BL21(DE3) (NEB) and purified as described previously[11,20,51]. Briefly, the protein was expressed in BL21(DE3)-competent cells overnight at 18 °C with 1 mM isopropyl β-d-1-thiogalactopyranoside. The cells were lysed in lysis buffer (20 mM MES, 100 mM NaCl, 2 mM EGTA, 2 mM EDTA, 5 mM dithiothreitol (DTT), pH 6.5) and the spin supernatant was precipitated with 40% ammonium sulfate before binding to a homemade cAMP-coupled agarose resin. The protein was eluted from the resin with cGMP (20 mM cGMP in lysis buffer) and run on a size-exclusion column to remove the excess cGMP. The protein is stored in gel filtration buffer (50 mM MES, 200 mM NaCl, 2 mM EGTA, 2 mM EDTA 5 mM DTT, pH 5.8)

**Attachment of dsDNA handles to protein constructs.** We followed a protocol previously published by the Maillard laboratory[14]. Briefly, the purified target protein was concentrated to ~5 mg/mL in 10 mM DTT to reduce all cysteine residues. The protein solution was run through three Micro Bio-Spin columns (Bio-Gel P6, Biorad) to remove the DTT before adding 10 mM 2,2′-dithiodipyriine (DTDP, Sigma) for 2 h at room temperature. The unreacted DTDP was removed from the modified protein using three additional Micro Bio-Spin columns. The DTDP-activated protein and two different 30 bp 5′-Thiol-modified dsDNA oligos were combined in a 1-to-1 molar ratio and incubated overnight at 4 °C. The resulting protein-oligo chimera was stored at −80 °C. Each 30 bp 5′-Thiol-modified dsDNA oligo used in the formation of the protein-oligo chimera has a unique non-palindromic overhang that is used to ligate 350 bp dsDNA handles modified with either biotin or digoxigenin in their 5′-end. The crosslinking reaction was done in the DNA crosslinking buffer (50 mM Tris, 100 mM NaCl, pH 7.6). Single-molecule optical tweezers measurements of protein constructs in the apo state were obtained by gradient elution (using cAMP) of the protein-oligo chimera from a cAMP-coupled agarose resin. The first eluted samples have an initial cAMP concentration of 20 μM, which is further diluted to a final concentration of ~0.02 nM (100-fold < $K_d$) before using it in the optical tweezers experiment.

**Optical tweezers measurements.** All data were collected in a MiniTweezers instrument[52]. Measurements were carried out in DNA crosslinking buffer. The protein sample with dsDNA handles were mixed with of 3.1 μm polystyrene beads (Spherotech) coated with anti-digoxigenin antibodies (termed AD bead) for 5 min at room temperature. The sample is diluted to 1 mL before applying it to the optical tweezers microchamber. The optical tweezer experiments were performed in DNA crosslinking buffer in a temperature-controlled room at 20 °C. A 2.1 μm bead coated with streptavidin (termed SA bead) is trapped on the tip of a micropipette, whereas the AD bead conjugated with the sample is trapped in the optical trap. To form a single tether, the AD bead in the optical trap was moved towards the SA beads on the micropipette tip. A single tether is confirmed by observing over-stretching of the DNA handles at ~65 pN[53].

Data were collected in two modes as follows. (1) Force-ramp experiments: to mechanically manipulate the target protein construct, we moved the AD bead in the optical trap away and towards the SA bead on the micropipette tip, which results in force-extension curves. The experiment was conducted at a constant pulling velocity of 75 nm s$^{-1}$, with a 10 s refolding time at 2 pN. Data were collected at a sampling rate of 200 Hz. For each protein construct in each experimental condition we collected 600–1200 trajectories from 6–12 different molecules. Rupture forces representing the cooperative unfolding of one or more protein domains and their associated extension changes were analyzed using a custom-built program implemented in MATLAB software. Unfolding force probability distributions obtained from force-extension curves were transformed to folded-state lifetimes as a function of force and analyzed using the methodology described by Dudko et al.[19] (full details are provided in the Supplementary Methods). (2) Force-clamp experiments: the protein was stretched and then held at a desired constant force using constant-force feedback loop. Changes in position at a particular force were recorded at a frequency of 500 Hz for the R241A mutant and 1000 Hz for the wild-type protein. The data from force-clamp experiments was analyzed using a BHMM approach[28,54].

**Molecular dynamics simulations.** MD and SMD simulations were performed with NAMD (v2.12)[55]. The CHARMM27 force field[56] was used for the protein and

counterions, and the TIP3P[57] for water. Parameters for cAMP were obtained with the CHARMM general force field[58]. The cAMP-bound state was modeled starting from the X-ray structure (PDB code: 1RGS)[11]. The apo state was obtained by removing cAMP from both the binding sites. Regulatory subunit of PKA with and without cAMP were solvated in a cubic box of 90 Å side. Systems included 24,200 water molecules and counterions were added to guarantee charge neutrality. The bonds between hydrogens and heavy atoms were constrained with the SHAKE algorithm[59]. The r-RESPA multiple time step method[60] was employed where long-range electrostatic interactions are updated every 4$fs$ and all the other interactions every 2$fs$. Periodic boundary conditions were used and the long-range electrostatics was treated with the particle-mesh-Ewald (PME) method[61] using a grid of 81 × 81 × 81. The cutoff for non-bonded interactions was set to 10 Å and the switching function was applied to smooth interactions between 9 and 10 Å. Simulations were conducted in the NPT (constant particle Number, Pressure and Temperature) ensemble. Temperature was set to 310 K through a Langevin thermostat[62] and pressure was set to 1 atm through a isotropic Langevin piston manostat[63]. The systems were first minimized (2000 steps of conjugate gradient) and equilibrated for 800 ps with the atoms of protein restrained to their initial positions. Production runs for both systems where 230 ns long. The holo structure at 90 ns was used to model the R241A mutant. The R241A mutant with cAMP was minimized and equilibrated as above and production runs was 200 ns long.

RMSD was evaluated over the Cα atoms considering all residues between 119 and 370, excluding the six residues of the unstructured N- and C-terminal tails. RMSD based clustering over Cα atoms was performed in the last 50 ns of simulation employing the hierarchical algorithm as implemented in Wordom[64] and a threshold of 2 Å (Supplementary Fig. 4a, b). The potential interaction energy, consisting of the sum of the Lennard-Jones term and the real part of the electrostatic interaction, between the protein domains was evaluated post processing the last 50 ns of the trajectory using the same parameters used to perform the standard simulations.

SMD[65] simulations were performed starting from a set of conformations from the production phase. In particular, four conformations for each cAMP-bound and apo states were selected from the last 50 ns of the simulation (Supplementary Fig. S4a). For each selected conformation, the protein was centered into the box and rotated to place the Cα atoms of the first and the last residue along the $x$-axis. To take in account the elongation of the protein during the pulling, the box was enlarged, by adding water molecules along the $x$ direction, by 170 Å. Finally, the box for SMD simulations was 260 × 90 × 90 Å$^3$ and the grid for the PME set to 256 × 81 × 81. The solvent was equilibrated for 400 ps harmonically restraining the protein in its original conformation and another 400 ps during which the restraints were progressively turned off. All SMD simulations were conducted in the NPT ensemble by using the same parameters employed to carry out the MD simulations. The final conformations were then used as starting point for the SMD simulations. The SMD simulations were performed by restraining the Cα atom of the first residue to its initial coordinates and applying the pulling constant velocity to a dummy atom attached via a virtual spring to the Cα atom of last residue: the spring constant was set to 0.5 kcal mol$^{-1}$ Å$^{-2}$ and the pulling velocity was set to 2.5 Å ns$^{-1}$ along the $x$ direction. Force as a function of the extension was averaged over each set of four SMD simulations. Clusters analysis was also performed for the four unfolding trajectories for each apo and cAMP-bound states. Snapshots of the trajectory where clustered, according to the end-to-end distances, are shown in Fig. 3b and Supplementary Fig. 4. The same SMD protocol was used also for the mutant R241A. However, as for R214A interactions between protein and cAMP are lost during equilibration, forced unfolding occurs similar to that in the apo state and only one simulation was performed.

**Reporting summary**. Further information on research design is available in the Nature Research Reporting Summary linked to this article.

## Data availability
The source data underlying Figs. 1a, 2a–d, 6d, h and 7c and Supplementary Figs. 1a and 5d are provided as a Source Data file. Other data are available from the corresponding author upon reasonable request.

## Code availability
All custom-made MATLAB codes used for the analysis of single-molecule unfolding and refolding trajectories are available from the corresponding author upon request.

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

## Acknowledgements

We thank Maria Fe Lanfranco, members of the Maillard lab, and Tsan-Wen Lu from the Taylor lab for constructive discussions on the manuscript. We thank Amy Chau from the Maillard lab for helping in collecting cAMP titration data for the isolated CNB domains. We thank Emília Pécora de Barros from the Amaro lab at UCSD for providing the atomic coordinates of the simulated R241A structure. We thank Joseph Lesniewski from the Jorabchi lab at Georgetown University and Alan Lowe from University College London for assisting in the global fitting code using PyFolding. This work was supported by NSF Grant MCB1715572 (R.A.M.), NIH Grants GM034921 (S.S.T), GM130389 (S.S.T.), R00CA187565 (H.C.H.), Cancer Prevention & Research Institute of Texas Grant RR170036 (H.C.H.), the V Foundation grant V2018–003 (H.C.H.), and Gabrielle's Angel Foundation for Cancer Research (H.C.H.).

## Author contributions

Y.H., S.S.T. and R.A.M. conceived and designed the research. Y.H. purified and modified all protein constructs. Y.H. and J.P.E. performed optical tweezers experiments. L.B. performed the molecular dynamic simulations. Y.H., H.C.H., and R.A.M. analyzed the data. Y.H., E.P., H.C.H., S.S.T. and R.A.M. wrote the manuscript.
