## [Peer Review File · Nature Communications]

Reviewers' comments:

Reviewer #1 (Remarks to the Author):

Hao and co-authors present a comprehensive and rigorous study of the effects of cAMP binding to the cyclic nucleotide binding domains of type I regulatory subunits of Protein Kinase A using optical tweezers. Studies applying this technique to signalling proteins are still a rarity, and certainly Protein Kinase A is an appropriate prototype protein to focus on. The key findings are that cAMP re-organises inter-domain interactions between the two CNBs, and that the N3A motif in particular changes conformation upon cAMP binding. I think this investigation will appeal to researchers interested in Protein Kinase A as well as structural biologists focused on cell signalling. It will encourage structural biologists to think beyond static conformations provided by crystallography, which still dominate structural thinking in cell signalling. The novelty is obviously lessened by the publication of a previous paper applying the same technique to the same protein (England et al., PNAS, 2018, PMID 30038016). However, the focus is different here (cAMP-induced dynamics rather than comparison +/- catalytic subunit), many different geometric cysteine pairings are applied, a wider range of optical tweezer approaches are employed, and the findings are more interesting. The techniques are clearly described and should be sufficient for other researchers hoping to apply similar approaches.

I have a number of minor comments/questions:

1. The title is generic. I appreciate that the authors are targeting a wide audience but there is no mention of Protein Kinase A, optical tweezers (only experimental technique used in the paper) or cyclic nucleotide binding domain. I worry that the study could be missed in literature searches but appreciate that it would be out of place to insist on a title change
2. Why is y-axis not labelled 0,5,10,15 in figure 1c?
3. On p. 16, the authors mention that two cysteine mutations (C345A, C360A) were introduced to prevent DTDP reacting at these sites. Is there a possibility that these mutations will have altered the structure/stability of CNB-B?
4. The language is too strong in places, e.g., in the abstract "the structural and dynamic features that enable the cyclic nucleotide binding signal to allosterically regulate other functional domains remain unknown". Many previous structural studies have addressed the same problem - it may not be resolved but unknown is too strong
5. On p. 10. Regarding dissociation constants: how do these values match up to dissociation constants determined by other approaches in previous studies?

Comments relating to the discussion. In general, I thought the discussion was too brief and did not sufficiently consider how the results matched up to mechanisms proposed in related studies. Specifically:

6. Is the 3CA motif likely to play a similar role in type II PKA regulatory subunits?
7. How does the mechanism put forward here compare to the movement of the N3A region proposed in the modelling study of Malmstrom et al., Nature Comms, 2015 (PMID 26145448)
8. Could potentially refer to kinetic studies – some of these emphasise that the cAMP-R-C complex is likely to be highly populated, and the mechanism here suggests a more stable conformation for this complex than put forward in previous studies (compare to tug-of-war models with direction competition between catalytic subunit and cAMP for the same elements in the regulatory subunit)
9. Is the stabilisation of the CNBs + cAMP observed here consistent with NMR studies, e.g., Das et al., PNAS, 2007, PMID 17182741?
10. There is no attempt to reconcile the findings of this study with the related 2018 study (England

et al., PNAS, 2018, PMID 30038016). Surely this should be attempted.

11. How does the proposed mechanism compare to the mechanisms put forward in studies applying crystallography, e.g., Kim et al, Science, 2005, PMID 15692043?

Reviewer #2 (Remarks to the Author):

Hao et al. investigated folding/unfolding processes of the regulatory cAMP-binding subunit of PKA using force spectroscopy and steered molecular dynamics. Information obtained was used to infer mechanisms underlying the allosteric regulation during PKA activation. Main results include the identification of the N-terminal cAMP-dependent dynamical switch, N3A, which was shown to be unstable upon cAMP binding due to enhanced domain-domain interactions. The authors further tested their hypothesis using mutagenesis and different nucleotide analogs.

The manuscript is overall well presented, and results are very interesting. I have questions related to the MD part:

- In experiments, the authors found that N3A was stable under the apo condition but became unstable and unfolded first upon cAMP binding. They then saw the same results in simulations. However, it is not mentioned how many simulations were performed for each condition and how often the same results were observed. From the text, it seems that only one trajectory was performed for each case. Multiple simulation replicates are required to validate robustness of observations. The same comment applies to the mutant simulation.

- Steered MD was used to test the stability of N3A under apo, cAMP-bound, and mutational conditions. A much simpler method is to directly compare the flexibility (e.g., RMSF) of N3A from regular MD after reaching equilibrium.

Reviewer #3 (Remarks to the Author):

Review of Hao et al, Nature Communications.

In their manuscript, "Activation of a Protein Kinase Via Asymmetric Allosteric Coupling of Structurally Conserved Signaling Modules" Yuxin Hao and co-workers describe single-molecule force-spectroscopic based analysis of the communication and allosteric interactions between the two cyclic nucleotide binding domains (CNBs) in protein kinase A. They employ optical tweezers probe the mechanical unfolding of the two cyclic nucleotide domains in several different contexts, in both apo and substrate bound conformations, and they introduce a point mutation to perturb a putative allosteric interaction between the two to regulatory cyclic nucleotide binding domains. Through elegant systematic measurements of the mechanical unfolding and careful analysis of the force distributions the authors provide a detailed free-energy description of the stabilization of each domain alone by the binding of the cyclic nucleotide (cAMP), the stabilization of each domain in the context of the other, and the full two-domain context. Their results demonstrate an asymmetric stabilization of the two CNB-domains by when bound to cAMP. Their data also reveal a novel domain associated with one of the CNB domains (CNB-A) that appears to be destabilized when the domain binds substrate. The authors follow up on the mechanistic relevance of this domain by mutating a key residue and testing the effect of this on the unfolding mechanics. They also perform an impressive series of measurement of the unfolding of the full construct over a titration of substrate cAMP, and in the presence of a weakly bound substrate, cGMP. From these data the authors construct a detailed map of communication and allostery between the two CNB domains associated with substrate binding. They postulate how these interactions and the intriguing destabilization of the small domain they identify can lead to activation of the PKA enzyme.

These are excellent experiments, well-conceived and well-executed to leverage the single-molecule force spectroscopy approaches to address fundamental questions concerning protein communication and allostery in multidomain substrate binding proteins. The data are of the highest quality and sufficient quantity for high precision measurements. The overall approach is systematic and thorough. The analytical approaches are equally as impressive. The results are

highly informative and set a paradigm for how to probe these kinds of systems with single-molecule approaches. Overall this is technically of very high quality. Despite the technical quality of the work, there are several points that the authors should address prior to acceptance of the manuscript. Most of these have to do with appropriately framing or placing their work in the context of the full PKA enzyme.

Main points

The N3A results are intriguing but it seems that the real test would be to determine the effect of the R241A mutation on the catalytic activity of the PKA enzyme. This would significantly bolster the claims surrounding the importance of this mutation on the conformational dynamics and communication between the two CNB domains. The single-molecule measurements are compelling but the connections to the full enzyme are tenuous throughout the manuscript, if this mutant had a predicted effect on PKA activity then the connection of the current work to the activity of the enzyme would be significantly enhanced. As it stands this remains a weak aspect of the work, despite its solid results.

In a similar vein, can the authors connect the results obtained with cGMP to known effects of cGMP on activation of PKA? These two perturbative measurements along with the cAMP titration results offer points of quantitative connection between the CNB domain measurements and the enzymatic activity of PKA.

The authors should clearly and succinctly place this work in context of their previous PNAS publication that covers similar but different ground. This will help place the current work in the broader context of their research effort and eliminate any concerns of overlap between the published and submitted work.

Finally, the relationship between the measured data and the simulations and how this plays into the final results should be clarified. I was impressed with the MC simulations but it remains somewhat unclear how exactly they are related to the data analysis.

Minor points / Details

Abstract: The abstract was not very clear and did a disservice to the elegant and thorough experiments and analysis. A few points: make it clear that PKA contains two regulatory CNB domains that synergistically bind substrate cAMP to activate PKA.

The sentence including "... time the pathways of signals transduced by cAMP binding in protein kinase A (PKA)" does not make sense. "Pathway" is too generic here.

"the folding energy landscape ... of PKA: Please be specific here: there is a great deal of confusion in the abstract concerning the experiments and how CNB relate to the PKA. The measurements were done with the two domains not the entire protein? If this is correct, then it should be stated in the abstract.

P 8. The N3A refolding experiments are a clever idea, but it is not clear that the refolding force of the N3A motif has been established. Without directly demonstrating that the N3A motif can refold at 5 pN the refolding experiments in which the CNB-B domain is unfolded are inconclusive since it is equally as likely that the N3A domain fails to refold on the timescale of the measurement at 5 pN.

Figure 4 is very confusing - it would be helpful to color code the contact map with the residues in each of the three domains; N3A, CNB-B and CNB-A domains. Likewise the cartoons on the right of Fig 4B should be better labeled or better described to make the domains clear.

P9. Figure 5a does not really relate to titration of cAMP between 1-150 nM – perhaps Figure 5 in general is related to this measurement but not specifically Figure 5a that only contains two measurements.

P10. The lack of folding or unfolding of the N3A domain between the partially cAMP bound CNB domains is not well-motivated in figure 5, or in the text. To show the lack of a folded N3A in fig 5C there should be an example trajectory showing what would be observed in the presence of a folded N3A domain – either experimental or computed. It is not entirely clear what the expected signal would be if the N3A domain were folded.

In Figure 5 it is unclear precisely how the different states of partial binding of cAMP were parsed. How could individual unfolding trajectories be assigned to one or the other states, of which there are a total of four? My impression is that this is based in part on the MC simulations, but this was not at all clear from the text or the figure caption.

P11. "In the presence of cAMP, however, the trajectories of R241A revealed an unfolding pathway that looked similar to that of wild type" this is confusing – the "however" would seem to indicate a

change of some sort – but the conclusion is that there is little apparent change.

Figure 6 a. it would be helpful to compare the unfolding trajectories with WT trajectories in this figure.

P12 the data supporting the claims of differences in the delta Lc between the R241A and Wt constructs should be presented. From Fig 6d there does not seem to be a significant difference, so this point needs to be clearly supported in the figure or in a supplemental figure.

The effects of the R241A mutation on PKA activity should be measured or cited. The authors have made a number of broad and detailed claims concerning the effects of the N3A motif on the binding and coordination of binding cAMP which are (mostly) supported by the single-molecule data but they need to establish that the effects that they observe result in changes in the activity of the full enzyme. Ideally, they should predict then verify their predictions of the effects of an N3A mutation on the activity of PKA but simply testing the mutant enzyme would be sufficient to demonstrate an effect – but some verification of the effects of the N3A mutation on enzyme activity is essential.

Figure 6 F – what do these cartoons represent? Are these results of simulations? Or are they representations of what the authors think is happening? Please provide the details in the figure legend.

P13 the statement " Negative coupling triggered by cAMP binding effectively melts interactions established between the N3A motif and the catalytic subunit, thereby facilitating the dissociation of the PKA complex" seems to come from nowhere – the work has largely focused on the interactions among the CNB domains and N3A – the interactions with the remainder of the PKA are speculative and have not been addressed elsewhere in the manuscript. These interactions could be estimated or examined via MD simulations of the N3A mutant, or through tests of the effect of the N3A mutation on the activity of PKA enzyme – speculation is okay in the discussion section but this seems to be somewhat disconnected from the remainder of the results and should be better motivated. And again, later in the conclusion, the authors propose a fairly detailed model of how their observations relate to the reorganization taking place in the PKA enzyme on cAMP binding – these are valuable insights but they should be better motivated.

Figure 8. It seems that an equivalent free energy diagram for the combined domains would be a good addition to the figure. The energy diagram for each individual CNB domain in the protein is useful, but the full combined energy diagram including the effects of N3A would make better connection with the full PKA enzyme and in conjunction with this diagram would highlight the cooperation between CNB domains.

Part B needs a great deal more explanation in the caption. What is PDE?

The pathway is constructed with a particular order of B binding cAMP first – is this established?

Can the alternative pathway be completely discounted?

Can the authors paint a picture – i.e., give an intuitive feel- for how the cycle in B results in activation of PKA – and how the R241A and cGMP alternative pathways would alter the activation – and crucially have these two effects been tested on enzyme activity?

P16 missing word? "...handles was mixed with of 3.1 μm..."

P19 Typo: "was applied to smoot interactions"

P33 the cAMP Kd values reported for the isolated CNB-A and CNB-B domains do not agree with the values reported in Figure 5 e.

P34 Typo missing words? "The BHMM analysis method have been previous described and applied in analyzing single molecule trajectories..."

Once these issues are addressed I would recommend publication of this elegant work.

Dear Reviewers:

We thank you for taking the time to carefully read our manuscript, and provide important suggestions, comments and critiques. The comments we received guided us to better connect our findings with previously published studies with PKA. Below you will find detailed responses to all major and minor comments. We hope that we were able to address your questions in a clear manner. All the new additions and modifications in the manuscript are outlined in blue font. We believe that the revised manuscript provides a clearer presentation of our data and analysis, as well as a deeper interpretation of our findings in the discussion section.

Reviewer #1

1. The title is generic. I appreciate that the authors are targeting a wide audience but there is no mention of Protein Kinase A, optical tweezers (only experimental technique used in the paper) or cyclic nucleotide binding domain. I worry that the study could be missed in literature searches but appreciate that it would be out of place to insist on a title change.

Thank you for your suggestion on a detailed title. **Please refer to our new title** “Activation of Protein Kinase A Via Asymmetric Allosteric Coupling of Two Structurally Conserved Cyclic-nucleotide Binding Domains”

2. Why is y-axis not labelled 0,5,10,15 in figure 1c?

Thank you for pointing it out. **Please refer to the revised figure 1c on pp. 56.**

3. On p. 16, the authors mention that two cysteine mutations (C345A, C360A) were introduced to prevent DTDP reacting at these sites. Is there a possibility that these mutations will have altered the structure/stability of CNB-B?

We would like to thank the reviewer for bringing out this point, **this information now has been added in the Methods part on pp. 21 in blue font**. From our previous publication (England, Jeniffer P., et al. "Switching of the folding-energy landscape governs the allosteric activation of protein kinase A." *Proceedings of the National Academy of Sciences* 115.32 (2018): E7478-E7485. <https://doi.org/10.1073/pnas.1802510115>), we found that the mutation C345A/C360A in CNB-B domain does not alter the solution structure, stability or the ability to form an inactive complex with the PKA catalytic subunit. In addition, compared to bulk studies, our single-molecule titration data for CNB-B domain showed a similar binding affinity constant for cAMP either as an isolated domain ($K_d = 27$ nM) or as part of the regulatory subunit ($K_d = 10$ nM), suggesting the binding affinity is not affected by the two cysteine mutations.

4. The language is too strong in places, e.g., in the abstract “the structural and dynamic features that enable the cyclic nucleotide binding signal to allosterically regulate other functional domains remain unknown”. Many previous structural studies have addressed the same problem - it may not be resolved but unknown is too strong.

We thank the reviewer's suggestion on the abstract. We have deleted the strong language, including the "unknown" word. **Please refer to our revised abstract.**

5. On p. 10. Regarding dissociation constants: how do these values match up to dissociation constants determined by other approaches in previous studies?

We thank the reviewer for suggesting the comparison of the values to other approaches in the previous studies. **Please refer to the revised paragraph on pp. 11 (bottom) -12 (top).** In our study, the single molecule titration and the extracted microscopic binding affinity for the isolated CNB-A and CNB-B domains are 84 nM and 27 nM, respectively. These values are consistent with previous studies of $EC_{50, \text{isolated CNB-A}} = 151 \text{ nM}$ and $EC_{50, \text{isolated CNB-B}} = 4 \text{ nM}$, measured by SPR (Lorenz, Robin, et al. "Mutations of PKA cyclic nucleotide-binding domains reveal novel aspects of cyclic nucleotide selectivity." *Biochemical Journal* 474.14 (2017): 2389-2403. doi: 10.1042/BCJ20160969).

For the regulatory subunit, we measured $K_{d, \text{CNB-A}} = 17 \text{ nM}$ and $K_{d, \text{CNB-B}} = 10 \text{ nM}$ as the first cAMP binding event. These data are comparable with previous bulk studies measured by [^3H] cAMP binding and exchange experiment ($K_{d, \text{CNB-A}} = 60 \text{ nM}$ and $K_{d, \text{CNB-B}} = 15 \text{ nM}$). We also showed that these values are consistent with a recent study with ITC, where $K_{d, \text{CNB-A}} = 4.2 \text{ nM}$ and $K_{d, \text{CNB-B}} = 2.8 \text{ nM}$ for each CNB domain in regulatory subunit (Poppe, Heiko, et al. "Cyclic nucleotide analogs as probes of signaling pathways." *Nature methods* 5.4 (2008): 277. <https://doi.org/10.1038/nmeth0408-277>).

Comments relating to the discussion. In general, I thought the discussion was too brief and did not sufficiently consider how the results matched up to mechanisms proposed in related studies. Specifically:

6. Is the N3A motif likely to play a similar role in type II PKA regulatory subunits?

This is an important point that we originally did not discuss in detail. **We have included this point in the discussion on the 1st paragraph on pp. 17:** "Given the remarkable structural similarity between different regulatory subunit isoforms bound to the catalytic subunit ..."

We proposed that the function of N3A motif in the activation is shared in other PKA isoforms, given the markable structural similarity between 0.5-0.6 Å between RI α -RII α (PDB 1QCS vs 2QVS), or RI α - RII β (PDB 1QCS vs 3TNQ). However, due to the specific interface between the catalytic subunit and each regulatory-subunit isoform in the heterotetramer (R $_2$ C $_2$), the N3A motif function might vary to different extents in the quaternary structure of PKA.

Referenced publications for RII α : Wu, Jian, et al. "PKA type II α holoenzyme reveals a combinatorial strategy for isoform diversity." *Science* 318.5848 (2007): 274-279. DOI: 10.1126/science.1146447. For RII β : Zhang, Ping, et al. "Structure and allostery of the PKA RII β tetrameric holoenzyme." *Science* 335.6069 (2012): 712-716. DOI: 10.1126/science.1213979.

7. How does the mechanism put forward here compare to the movement of the N3A region proposed in the modelling study of Malmstrom et al., Nature Comms, 2015 (PMID 26145448)

We have incorporated this work with our model in the detailed discussion on the 2nd paragraph (middle) of pp. 16. “The conformational change of the B/C helix may propagate to the N3A motif in the CNB-A domain ...”. We believe that, by including Malmstrom’s work, our description on the proposed model is more detailed and clear. In the model proposed by Malmstrom et al, the B/C helix propagates the binding signal to the N3A motif, and to the interface of the catalytic subunit. It also described a substantial movement of N3A motif in the activation of PKA. This model reconciles the mechanism we put forward in our findings: the cAMP-binding signal from the CNB-B domain propagate to the B/C helix, and allosterically to the N3A motif in CNB-A domain. Since both the B/C helix and N3A motif in CNB-A domain make the majority contacts in the surface interface between the regulatory subunit and the catalytic subunit, the gradual dissociation of these interactions will prompt the activation of PKA.

8. Could potentially refer to kinetic studies – some of these emphasize that the cAMP-R-C complex is likely to be highly populated, and the mechanism here suggests a more stable conformation for this complex than put forward in previous studies (compare to tug-of-war models with direction competition between catalytic subunit and cAMP for the same elements in the regulatory subunit)

The tug-of-war models for PKA has been previously described by Das et al. (Das, Rahul, et al. "cAMP activation of PKA defines an ancient signaling mechanism." *Proceedings of the National Academy of Sciences* 104.1 (2007): 93-98. <https://doi.org/10.1073/pnas.0609033103>). We believe this model also applies to our proposed mechanism: both catalytic subunit and cAMP are competing for the regulatory subunit, especially in the N3A motif. The N3A motif makes very distinct contacts depending on cAMP or catalytic subunit interactions. A hybrid state (regulatory-catalytic complex with partial liganded cAMP) has been proposed to exist in the activation process of PKA.

Our thermodynamic studies have suggested that the cAMP-bound regulatory subunit is more stable, thereby leading the success in the competition and resulting in the release of the catalytic subunit. **Please refer to our newly added discussion on 2nd paragraph (bottom) of pp. 19:** “Thus, the catalytic subunit and cAMP are competing for similar interaction regions in the regulatory subunit ...”.

9. Is the stabilization of the CNBs + cAMP observed here consistent with NMR studies, e.g., Das et al., PNAS, 2007, PMID 17182741?

The stabilization of cAMP to CNB domain described in previous studies are 9 ~10 kcal/mol. (Das, Rahul, et al. "cAMP activation of PKA defines an ancient signaling mechanism." *Proceedings of the National Academy of Sciences* 104.1 (2007): 93-98. <https://doi.org/10.1073/pnas.0609033103> and Cànaves, Jaume M., Darryl A. Leon, and Susan

S. Taylor. "Consequences of cAMP-binding site mutations on the structural stability of the type I regulatory subunit of cAMP-dependent protein kinase." *Biochemistry* 39.49 (2000): 15022-15031.) doi: 10.1074/jbc.272.26.16343)

We calculated the cAMP stabilization to the regulatory subunit from the single molecule data to be 12.5 kcal/mol, which is the combination of the cAMP-induced domain-specific stabilization and inter-domain interaction. With optical tweezers data we showed here additional insights on stabilization of each domain by selectively manipulation: CNB-A is stabilized more than CNB-B domain which can be attributed to the N3A motif. **We have included this comparison on pp. 6 (bottom)-7 (top)** “cAMP binding stabilizes the CNB-B domain from 7.6 kcal/mol to 10.4 kcal/mol ...” **and 3rd paragraph (middle) of pp. 15** “The thermodynamic stability of the isolated CNB domains bound to cAMP is comparable to ...”

10. There is no attempt to reconcile the findings of this study with the related 2018 study (England et al., PNAS, 2018, PMID 30038016). Surely this should be attempted.

Thank you for suggesting reconciling this study with previously published paper in our group. **We have addressed this point on 1st paragraph of pp. 19** “In a recent study we showed how the CNB domains of the regulatory subunit (RI α) are thermodynamically ...”

While the previous work showed how the CNB domains are thermodynamically poised in such a way that CNB-B controls the stability of CNB-A, providing a foundation on why cAMP binding to the B domain triggers allosteric networks of communication to the A domain. In this study, we directly interrogate the forces involved in driving the cAMP-mediated activation of PKA. Mechanical fingerprints are unique for each state either R-C (regulatory and catalytic subunit) or R-cAMP2 (one regulatory subunit bound to 2 cAMP molecules). These unique fingerprints may emerge from the fact that these signaling modules have evolved to behave as molecular switches, changing in conformation or dynamic properties depending on the binding partner.

11. How does the proposed mechanism compare to the mechanisms put forward in studies applying crystallography, e.g., Kim et al, Science, 2005, PMID 15692043?

We have addressed the specific contacts that involved the dissociation between regulatory and catalytic subunit. **Please refer to the paragraph on 2nd paragraph (middle) of pp. 16** “In those studies, it is proposed that the binding of the first cAMP molecule to the CNB-B domain ...”

Combined with information from crystallographic studies, we proposed that the binding of the first cAMP to CNB-B domain induces a substantial conformational change in the B/C helix, breaking hydrogen bonding between Y247^C and Y205^R, together with a series of van der Waals interactions between B/C helix and the catalytic subunit. The motion of B/C helix can propagate the binding signal to N3A motif, further breaking the interaction between L135^R and Y224^C, I210^C, L204^C, resulting in a substantial movement of N3A motif and ultimately the unleash of

the catalytic subunit (The subscript notes the residue for regulatory subunit and catalytic subunit)

Reviewer #2

1. In experiments, the authors found that N3A was stable under the apo condition but became unstable and unfolded first upon cAMP binding. They then saw the same results in simulations. However, it is not mentioned how many simulations were performed for each condition and how often the same results were observed. From the text, it seems that only one trajectory was performed for each case. Multiple simulation replicates are required to validate robustness of observations. The same comment applies to the mutant simulation.

Thank you for pointing out the missing information about the simulation methods. We have provided a more detailed simulation method. **Please refer to our revision on pp. 25-26.** For wild type, four simulations were performed, and they all converged. Populated were derived from cluster analysis of all MDs, for both cAMP-bound and apo. Clusters analysis was also performed for the four unfolding trajectories for each apo and cAMP-bound states. The distributions of the forces with respect to the distance are from the average of the SMDs for both systems. Because R214A interactions between protein and cAMP are lost during equilibration, forced unfolding occurs like in the apo state and only one simulation was performed.

2. Steered MD was used to test the stability of N3A under apo, cAMP-bound, and mutational conditions. A much simpler method is to directly compare the flexibility (e.g., RMSF) of N3A from regular MD after reaching equilibrium.

We thank the reviewer for providing this important insight and future studies will be focus on both regular MD and SMD. Our original motivation was to directly compare single molecule unfolding experiments with SMD. Specifically, we were focus on the dynamics of unfolding of the N3A motif under force, and the consequences of inter-domain interactions.

Reviewer #3

1. The N3A results are intriguing but it seems that the real test would be to determine the effect of the R241A mutation on the catalytic activity of the PKA enzyme. This would significantly bolster the claims surrounding the importance of this mutation on the conformational dynamics and communication between the two CNB domains. The single-molecule measurements are compelling but the connections to the full enzyme are tenuous throughout the manuscript, if this mutant had a predicted effect on PKA activity then the connection of the current work to the activity of the enzyme would be significantly enhanced. As it stands this remains a weak aspect of the work, despite its solid results.

The reviewer brings an important point that we did not discuss in our original manuscript. **We included the discussion of N3A motif of R241 in terms of the catalytic activity of PKA on**

pp. 13 in blue font.

The R241A mutation we described here does not perturb the integrity or the binding affinity of each domain (P. Barros, Emília, et al. "Electrostatic interactions as mediators in the allosteric activation of protein kinase A RI α ." *Biochemistry* 56.10 (2017): 1536-1545. <https://doi.org/10.1021/acs.biochem.6b01152> and SI figure 6), bulk studies have shown that in order to activate the catalytic subunit, the R241A requires 20 folds more cAMP compared to wild type. Our finding here suggested that, due to a disruption of the conformational dynamics of the N3A motif, its ability to serve as an efficient cAMP-responsive molecular switch is largely diminished in the R241A mutant. This results in the impeding of the regulatory subunit to attain its final cAMP-bound conformation, and thereby leads to the insufficient activation of the PKA.

2. In a similar vein, can the authors connect the results obtained with cGMP to known effects of cGMP on activation of PKA? These two perturbative measurements along with the cAMP titration results offer points of quantitative connection between the CNB domain measurements and the enzymatic activity of PKA.

We thank the reviewer for the suggestions to connect the cGMP effects on the activation of the regulatory subunit of PKA. **We have added the comparison between cAMP and cGMP in the binding affinity of the regulatory subunit, and its activation constant for PKA holoenzyme on pp. 14 in blue font.** Briefly, previous studies showed that cGMP binding to PKA requires a higher concentration and results in lower cooperativity. The activation of PKA holoenzyme by cGMP is >140 fold less sensitive to cAMP-stimulated activation. Our single molecule studies in addition provide experimental evidence that nucleotide selectivity not only involves previously described defects in binding affinities and activation constants, but also an attenuation of inter-domain interactions and decoupling of cyclic nucleotide binding from the conformational switching of the N3A motif.

3. The authors should clearly and succinctly place this work in context of their previous PNAS publication that covers similar but different ground. This will help place the current work in the broader context of their research effort and eliminate any concerns of overlap between the published and submitted work.

Thank you for pointing out that the connection between PNAS work and this work should be made. In fact, another reviewer had the same question. Please refer to our answer to **Reviewer #1 in question 10** and also see **on 1st paragraph of pp. 19** "In a recent study we showed how the CNB domains of the regulatory subunit (RI α) are thermodynamically ..."

4. Finally, the relationship between the measured data and the simulations and how this plays into the final results should be clarified. I was impressed with the MC simulations but it remains somewhat unclear how exactly they are related to the data analysis.

We thank the reviewer for the comments regarding the simulation part. Another reviewer also

brought the same question. Please refer our answers to **Reviewer # 2 in question 1, 2**. And also, we have added a more detailed experimental description in **on pp. 25-26**.

Minor points / Details

5. Abstract: The abstract was not very clear and did a disservice to the elegant and thorough experiments and analysis. A few points: make it clear that PKA contains two regulatory CNB domains that synergistically bind substrate cAMP to activate PKA. The sentence including "... time the pathways of signals transduced by cAMP binding in protein kinase A (PKA)" does not make sense. "Pathway" is too generic here. "the folding energy landscape ... of PKA: Please be specific here: there is a great deal of confusion in the abstract concerning the experiments and how CNB relate to the PKA. The measurements were done with the two domains not the entire protein? If this is correct, then it should be stated in the abstract.

We thank the reviewer for the suggestions on the abstract. We did drastic changes in the abstract compared to the one we had originally. **Please see our updated abstract.**

1) We have stated that the regulatory subunit has two CNB domains "... to monitor the signals transduced by cAMP binding between the two CNB domains of the regulatory subunit of protein kinase A (PKA)." 2) We deleted the "pathway" and replaced with "monitor the signals transduced by cAMP binding" to be more specific. 3) We clarified that we did our experiment in the regulatory subunit of PKA by stating "We use optical tweezers and molecular dynamics to monitor the signals transduced by cAMP binding between the two CNB domains of the regulatory subunit of protein kinase A (PKA)." We also tried to make the connection between the folding energy landscape of CNB domain to the contribution of each domain in the activation of PKA", which can be referred to the sentences "We find that the response of the folding energy landscape to cAMP is ... during the allosteric activation of PKA."

6. The N3A refolding experiments are a clever idea, but it is not clear that the refolding force of the N3A motif has been established. Without directly demonstrating that the N3A motif can refold at 5 pN the refolding experiments in which the CNB-B domain is unfolded are inconclusive since it is equally as likely that the N3A domain fails to refold on the timescale of the measurement at 5 pN.

The reversible hopping signature of the N3A motif shows that it can dynamically oscillate between unfolded and folded states. Such dynamic behavior occurs between 10-12 pN, indicating that N3A motif is mostly unfolded at 12pN and folded at 10 pN. We would like to thank the reviewer for bringing up this question and **we clarified this point on the 3rd paragraph (in blue font) of pp. 9.**

7. Figure 4 is very confusing - it would be helpful to color code the contact map with the residues in each of the three domains; N3A, CNB-B and CNB-A domains. Likewise the cartoons on the right of Fig 4B should be better labeled or better described to make the domains clear.

We thank the reviewer for the suggestion on Figure 4. **Both figure panels in 4b (pp. 59) were updated with color-coded residues** (N3A motif: yellow; CNB-A domain: light purple; CNB-B domain: blue; B/C helix: red) and the figure legends were revised.

8. Figure 5a does not really relate to titration of cAMP between 1-150 nM – perhaps Figure 5, in general, is related to this measurement but not specifically Figure 5a that only contains two measurements.

We thank the reviewer for pointing out the possible confusion in Figure 5a. Indeed, the two trajectories showed in 5a are two representative trajectories for two intermediate-liganded states (A_1B_0 and A_0B_1) selected from the titration of cAMP. **Please refer to our revised figure legends for 5a on pp. 60.**

9. The lack of folding or unfolding of the N3A domain between the partially cAMP bound CNB domains is not well-motivated in figure 5, or in the text. To show the lack of a folded N3A in fig 5C there should be an example trajectory showing what would be observed in the presence of a folded N3A domain – either experimental or computed. It is not entirely clear what the expected signal would be if the N3A domain were folded.

Thank you for your comments on the clarity of Figure 5c. We have added an expected worm-are shown for CNB-A domain without the N3A motif (dark purple) and with the N3A motif (light purple), showing that the N3A motif in CNB-A domain in A_1B_0 become unfolded. We also incorporated an overlay of trajectories of the doubly-cAMP bound regulatory subunit that shows N3A motif folded (gray) and compared with the A_1B_0 that N3A motif becomes unfolded (red). **Please refer to our revised figure legends for 5a and 5c on pp. 60.**

10. In Figure 5 it is unclear precisely how the different states of partial binding of cAMP were parsed. How could individual unfolding trajectories be assigned to one or the other states, of which there a total of four? My impression is that this is based in part on the MC simulations, but this was not at all clear from the text or the figure caption.

We have provided a **detailed analysis for the four different states in supplementary information on pp. 44 in blue font.**

11. “In the presence of cAMP, however, the trajectories of R241A revealed an unfolding pathway that looked similar to that of wild type” this is confusing – the “however” would seem to indicate a change of some sort – but the conclusion is that there is little apparent change.

The reviewer is correct. **We changed this sentence** as “In the presence of cAMP the trajectories of R241A revealed an unfolding pathway that looked similar to that of wild type, but with some important quantitative differences” **on pp. 12.**

12. Figure 6 a. it would be helpful to compare the unfolding trajectories with WT trajectories in this

figure.

Since we have the wild type trajectories displayed in the main Figure 3, we omit it in the main Figure 6a. However, we did provide a direct comparison **in Supplementary Figure 6b with a zoom-in the trajectories.**

13. The data supporting the claims of differences in the delta Lc between the R241A and Wt constructs should be presented. From Fig 6d there does not seem to be a significant difference, so this point needs to be clearly supported in the figure or in a supplemental figure.

Thank you for pointing out the missed presentation for the data in the original manuscript to support our findings for N3A motif confirmation in R241A. **We have added a new panel in Figure 6f (pp. 61)** for the distribution of the change in contour length of the N3A motif or wild type and R241A, supporting our claim that the N3A motif undergoes a shorter change in contour length.

14. The effects of the R241A mutation on PKA activity should be measured or cited.

Yes. We have added the R241A mutation on the PKA activity and binding affinity. **Please refer to pp. 13 in blue font.**

15. The authors have made a number of broad and detailed claims concerning the effects of the N3A motif on the binding and coordination of binding cAMP which are (mostly) supported by the single-molecule data but they need to establish that the effects that they observe result in changes in the activity of the full enzyme. Ideally, they should predict then verify their predictions of the effects of an N3A mutation on the activity of PKA but simply testing the mutant enzyme would be sufficient to demonstrate an effect – but some verification of the effects of the N3A mutation on enzyme activity is essential.

We thank the reviewer for leading us to discuss the importance of N3A motif in the activity of PKA. Previous literature has identified key residues in the N3A motif region that critical for PKA activation: For example, mutations K121A or Y120A in N3A motif have been shown to decrease the Hill coefficient of activation of the PKA hetero-tetramer from 1.7 to ~ 1.0. The mutations S145G and R144S also located in the N3A motif and that are related to Carney Complex disease have a lower Hill coefficient of 1.4 (Bruystens, Jessica GH, et al. "PKA RI α homodimer structure reveals an intermolecular interface with implications for cooperative cAMP binding and Carney complex disease." *Structure* 22.1 (2014): 59-69. doi: 10.1016/j.str.2013.10.012). These mutations support our claims that N3A motif is a critical element not only in the communication between domains, also in the activity of PKA enzyme. **We incorporated this discussion on 2nd paragraph (middle) of pp. 18 “Consistent with this notion ...”**

16. Figure 6 F – what do these cartoons represent? Are these results of simulations? Or are they representations of what the authors think is happening? Please provide the details in the figure legend.

Thank you for pointing this out. The R241A is the simulated structure while the WT is the crystal structure from PDB 1RGS. **The figure legends of 6g is revised on pp. 61 of the single manuscript file.**

17. P13 the statement "Negative coupling triggered by cAMP binding effectively melts interactions established between the N3A motif and the catalytic subunit, thereby facilitating the dissociation of the PKA complex" seems to come from nowhere – the work has largely focused on the interactions among the CNB domains and N3A – the interactions with the remainder of the PKA are speculative and have not been addressed elsewhere in the manuscript. These interactions could be estimated or examined via MD simulations of the N3A mutant, or through tests of the effect of the N3A mutation on the activity of PKA enzyme – speculation is okay in the discussion section but this seems to be somewhat disconnected from the remainder of the results and should be better motivated. And again, later in the conclusion, the authors propose a fairly detailed model of how their observations relate to the reorganization taking place in the PKA enzyme on the cAMP binding – these are valuable insights but they should be better motivated.

We thank the reviewer for the suggestion to better motivate different aspects of our proposed model. In the revised manuscript **we provide more details to better describe the mechanism on pp. 17. In general, with the revisions suggested from all reviewers, we extend our discussion to a much broader context**, including a detailed proposed PKA activation model, and the importance of the N3A motif in the activity of PKA.

18. Figure 8. It seems that an equivalent free energy diagram for the combined domains would be a good addition to the figure. The energy diagram for each individual CNB domain in the protein is useful, but the full combined energy diagram including the effects of N3A would make better connection with the full PKA enzyme and in conjunction with this diagram would highlight the cooperation between CNB domains.

Thank you for the suggestion on a combined energy diagram as a conclusion figure. **Please refer to our newly added figure 8a on pp. 63.** We believe this combined energy diagram summarizes our most important findings in our work.

Similarly, the original folding landscapes from the individual domains (manipulated as isolated structures or selectively in the regulatory subunit) were moved at the end of Fig. 2 (figure legend revised). This modification has helped to better group one set of experiment and one set of results.

Part B needs a great deal more explanation in the caption. What is PDE?

We have added a detailed explanation in Figure 8b now on pp. 63.

19. The pathway is constructed with a particular order of B binding cAMP first – is this established?

Can the alternative pathway be completely discounted?

We thank the reviewer for pointing out this question. Both the CNB-A domain and the B/C helix make extensive surface contacts with the catalytic subunit, and the catalytic subunit blocks the CNB-A binding site. Therefore, cAMP has a preferential access to the CNB-B domain, behaving as a “gatekeeper”. (Kim, Choel, et al. "PKA-I holoenzyme structure reveals a mechanism for cAMP-dependent activation." *Cell* 130.6 (2007): 1032-1043. DOI: 10.1016/j.cell.2007.07.018 and Kim, Choel, Nguyen-Huu Xuong, and Susan S. Taylor. "Crystal structure of a complex between the catalytic and regulatory (RI α) subunits of PKA." *Science* 307.5710 (2005): 690-696. doi: 10.1016/j.str.2010.08.013). In our study, we find that the CNB-B domain has a higher cAMP binding affinity than the CNB-A domain. Our single molecule binding studies together with previous structural work motivated to construct the activation pathway where CNB-B binds to cAMP first.

20. Can the authors paint a picture – i.e., give an intuitive feel- for how the cycle in B results in activation of PKA – and how the R241A and cGMP alternative pathways would alter the activation – and crucially have these two effects been tested on enzyme activity?

Yes. **Please refer to the revised Figure 8b on pp. 63**, where we depict the activation mechanism of PKA and how R241A and cGMP disrupted the pathway.

21. P16 missing word? “...handles was mixed with of 3.1 μm ...”

You are correct. There should be no “of”

22. P19 Typo: “was applied to smoot interactions”

We changed it to “smooth interactions”

23. P33 the cAMP K_d values reported for the isolated CNB-A and CNB-B domains do not agree with the values reported in Figure 5 e.

Figure 5e showed the dissociation constant (K_d). While the value reported in the supplementary information is originally reported from the fitting software as association constant (k_a).

24. P34 Typo missing words? “The BHMM analysis method have been previously described and applied in analyzing single molecule trajectories...”

You are correct. We edited this sentence.